# Forgetting Order of Continual Learning: What is Learned First is Forgotten Last

## Abstract

Catastrophic forgetting poses a significant challenge in continual learning, where models often forget previous tasks when trained on new data. Our empirical analysis reveals a strong correlation between catastrophic forgetting and the learning speed of examples: examples learned early are rarely forgotten, while those learned later are more susceptible to forgetting. We demonstrate that replay-based continual learning methods can leverage this phenomenon by focusing on mid-learned examples for rehearsal. We introduce Goldilocks, a novel replay buffer sampling method that filters out examples learned too quickly or too slowly, keeping those learned at an intermediate speed. Goldilocks improves existing continual learning algorithms, leading to state-of-the-art performance across several image classification tasks.

## 1 Introduction

The surge in deploying deep models in real-world scenarios has highlighted the need for models capable of adjusting to changing environments. Continual learning (see extensive reviews, e.g (De Lange et al., 2021; Hadsell et al., 2020; Parisi et al., 2019)), which involves training deep learning models on new data even after mastering previous tasks, is crucial in this context. However, continual learning encounters a significant obstacle called catastrophic forgetting (French, 1999; Kemker et al., 2018; McCloskey & Cohen, 1989): as models learn from new data, they tend to forget previous tasks, resulting in decreased performance on those tasks. Despite extensive research efforts, catastrophic forgetting persists as a central challenge in continual learning.

In this work, we begin by empirically examining catastrophic forgetting, identifying which examples deep models tend to forget when trained on new tasks. Our findings reveal a last-in-first-out pattern, where examples learned later are forgotten more readily than those learned earlier. Fig. 1 visualizes this by showing which examples are likely to be forgotten or remembered over time. This phenomenon is connected to the simplicity bias (Shah et al., 2020; Szegedy et al., 2014) of neural networks, where networks learn simple examples faster than complex ones. However, complex examples are more likely to be forgotten, while simple examples are more likely to be remembered.

This observation has an important consequence: by measuring how fast examples are being learned by the model, we can predict in advance which examples are more susceptible to forgetting, and treat them accordingly. This holds true regardless of the distribution of the new task introduced to the model. Furthermore, we demonstrate that models showing varying degrees of forgetting do so consistently: models with less catastrophic forgetting recall most of the examples retained by those with more forgetting, with the additional ability to remember slightly more complex examples.

Replay-based continual learning methods use a memory buffer to address catastrophic forgetting, storing a small subset of the original task examples. Typically, this buffer is uniformly sampled from previous tasks, and different continual learning techniques vary in how they utilize it. Our findings indicate that while a replay buffer helps mitigate forgetting, a similar pattern emerges: examples learned later are forgotten sooner. Increasing buffer size or improving the underlying model's architecture can alleviate more of the forgetting to some extent, but the fundamental phenomenon persists. This observation implies that when strengthening a continually trained model, we can predict in advance which examples it will better remember compared to its weaker version.

This raises the question: can one exploit this phenomenon and reduce forgetting by cleverly selecting examples for the replay buffer? Our observations reveal that the model benefits most from examples

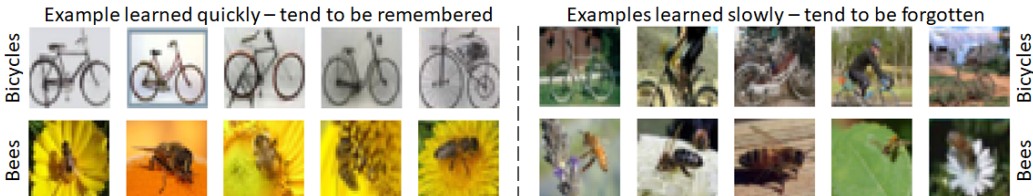

Figure 1: Test examples that networks tend to either remember or forget. Networks were trained on 2 tasks from CIFAR-100. Examples from the first task that were learned quickest (left) and slowest (right) are shown. Slowest examples were often forgotten, while the quickest were seldom forgotten.

it usually does not remember but still has the potential to. Given the fixed buffer size, these examples should come at the expense of the fastest-to-learn examples, which the model is likely to remember already, and the slowest-to-learn examples, which the model has less potential to remember. Leveraging these insights, we introduce a practical sampling method named Goldilocks for the replay buffer. When constructing the replay buffer, Goldilocks eliminates examples learned too quickly or too slowly, ensuring the buffer comprises examples learned at just the right speed.

Goldilocks is very cheap both computationally and memory-wise, requiring only a single float per example, which can be computed during the network's forward pass. While Goldilocks sampling improves the performance of the vanilla replay-buffer-based continual learning, it is also compatible with other continual learning methods. Since it only modifies the sampling function for selecting buffer examples, the rest of the continual learning method, which is often concerned with buffer utilization, remains unchanged. Goldilocks consistently improves the performance of all methods tested, achieving state-of-the-art results in many classification tasks.

## 1.1 Our contribution

- **Analysis of catastrophic forgetting**: We reveal a strong correlation between how quickly examples are learned by the model and their likelihood of being forgotten. This insight deepens our understanding of catastrophic forgetting, allows us to identify patterns of forgotten examples across different scenarios, and enables the prediction of which examples are more likely to be forgotten in advance.

- **Introduction of Goldilocks**: We propose Goldilocks, a novel and efficient replay buffer sampling method. Goldilocks selectively retains examples learned midway through training, discarding those that are either too simple or too complex. This approach is easily combined with many other continual learning methods, improving their performance and advancing the state of the art.

## 1.2 Related work

**Catastrophic forgetting.** Existing studies primarily focus on mitigating catastrophic forgetting (Kirkpatrick et al., 2017; Lee et al., 2017; Li et al., 2019; Ritter et al., 2018; Serra et al., 2018), rather than its characterization. Some works have explored catastrophic forgetting from a model perspective: (Nguyen et al., 2020; Ramasesh et al., 2021) showed differences in forgetting patterns across model layers, while (Mirzadeh et al., 2020; Pfülb & Gepperth, 2019) demonstrated the impact of training hyper-parameters. (Nguyen et al., 2019) identified task complexity as a factor influencing forgetting rates. Our study differs by examining the relationship between learning speed and forgettability of examples, emphasizing a data-centric perspective.

**Simplicity bias.** Empirical evidence supports the hypothesis that neural networks initially learn simple models, gaining complexity over time (Cao et al., 2021; Gissin et al., 2020; Gunasekar et al., 2018; Heckel & Soltanolkotabi, 2020; Hu et al., 2020; Jin & Montúfar, 2023; Kalimeris et al., 2019; Pérez et al., 2019; Soudry et al., 2018; Ulyanov et al., 2018). This phenomenon, known as simplicity bias (Dingle et al., 2018; Shah et al., 2020), is extended in our work to catastrophic forgetting, illustrating a reverse simplicity bias where complex examples are forgotten before simpler ones.

**Forgetting dynamics.** Several studies have directly explored the dynamics of forgetting. (Maini et al., 2022; Toneva et al., 2019) revealed that, even in non-continual settings, certain examples tend to be less forgettable during training. (Millunzi et al., 2023) demonstrated that the forgetting dynamics of noisy examples differ from those of clean labels and proposed adjusting the rehearsal process accordingly. Our results align with these findings, further delineating which examples are more prone to forgetting and retention, establishing connections between catastrophic forgetting and simplicity bias, and proposing a method to identify examples likely to be forgotten proactively. Additionally, our work illustrates how to leverage these insights to enhance the sampling of the replay buffer for various continual learning approaches.

**Replay buffer sampling functions.** Replay-based methods mitigate catastrophic forgetting by storing examples from previous tasks in a memory buffer of fixed size. Despite the availability of various sampling strategies (Aljundi et al., 2019; Benkő, 2024; Buzzega et al., 2021; Rebuffi et al., 2017; Tiwari et al., 2022; Wiewel & Yang, 2021), their effectiveness often depends on specific conditions or algorithms. As a result, many replay-based methods resort to uniform sampling (Buzzega et al., 2020; Guo et al., 2020; Kirkpatrick et al., 2017; Lopez-Paz & Ranzato, 2017; Prabhu et al., 2020; Ramesh & Chaudhari, 2022; Rolnick et al., 2019) in practice. Unlike previous methods, Goldilocks can improve replay-based methods across diverse tasks and scenarios.

## 2 EMPIRICAL ANALYSIS OF CATASTROPHIC FORGETTING

In this section, we empirically study catastrophic forgetting from a data-centric perspective, drawing a connection between the speed at which a model learns examples, and its likelihood to forget them.

### 2.1 DEFINITIONS

We examine the continual learning classification setting, where the training dataset $\mathcal{D} = \{\mathcal{X}, \mathcal{Y}\}$ consists of examples $x \in \mathcal{X}$ and their corresponding labels $y \in \mathcal{Y}$. This dataset is divided into $T > 1$ tasks, with each task $t \in [T]$ represented as $\mathcal{D}_t = \{\mathcal{X}_t, \mathcal{Y}_t\}$, and $\mathcal{D} = \bigcup_{t=1}^{T} D_t$. There is no overlap between the data of different tasks. A deep model $f : \mathcal{X} \rightarrow \mathcal{Y}$ is sequentially trained on each task $\mathcal{D}_t$ from $t = 1$ to $t = T$, with $E \in \mathbb{N}$ epochs per task. Each task also has a separate test set from the same distribution. During training on a task $\mathcal{D}_t$, the model $f$ is provided with a replay buffer $B$ of fixed-memory size, containing examples from previous tasks $1, ..., t - 1$. Notably, $|B| \ll |\mathcal{D}|$, meaning only a fraction of the data can be stored in the replay buffer. It's worth mentioning that the buffer may be empty ($B = \emptyset$), representing training $f$ without a memory buffer.

**Learning speed.** Neural networks demonstrate a trend in learning examples, where certain examples are consistently learned before others across various deep neural models (Baldock et al., 2021; Choshen et al., 2022; Hacohen et al., 2020). This orderliness is often connected to the simplicity bias phenomenon (Hacohen & Weinshall, 2022), as examples learned earlier tend to be simpler than those learned later. Rather than measuring the example's complexity, which can be defined in many different ways, we use the speed at which examples are learned by the deep model, which can be measured efficiently and directly. Given a model $f$ and an example $(x, y)$ either in the train or test data, we define the *learning speed* of the example $(x, y)$ to be:

$$learning\_speed(x, y) = \frac{1}{E} \sum_{e=1}^{E} \mathbb{1}[f_t^e(x) = y] \qquad (1)$$

Here, $f_t^e$ denotes the intermediate model $f$ after training for $e \in [E]$ epochs on task $\mathcal{D}_t$. This definition is based on the *accessibility score* introduced by Hacohen et al. (2020), which we adapt for computation using a single model instead of an ensemble, facilitating efficient evaluation throughout the training of $f$. Intuitively, this score correlates with how quickly the model learns an example: if an example is correctly classified from the early stages of learning, it will sustain $f_t^e(x) = y$ for more epochs, resulting in a higher *learning speed*. A discussion of alternative metrics to learning speed can be found in App. G. A detailed explanation of the relationship between the example's learning speed and this score is provided in Hacohen et al. (2020), and thus is not repeated here.

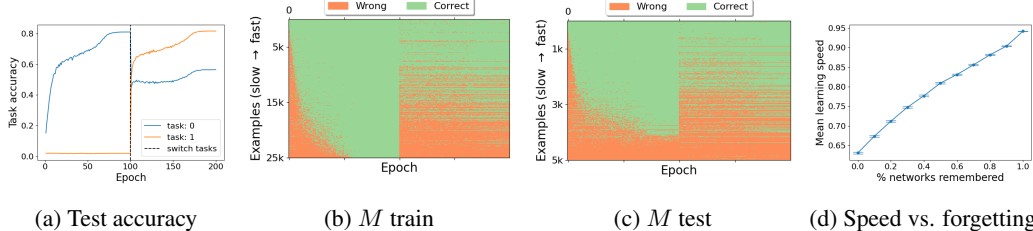

(a) Test accuracy        (b) $M$ train        (c) $M$ test        (d) Speed vs. forgetting

Figure 2: Forgetting as a function of the learning speed. We trained 10 networks on CIFAR-100-2, without a replay buffer. (a) Mean test accuracy of each task, where the dashed line marks the task switch. The models forget much of the first task after the switch. (b-c), the first task's binary epoch-wise classification matrices $M$ for train and test data. The y-axis corresponds to different examples, and the x-axis corresponds to different epochs, showing if the examples were classified correctly in each epoch. The order in which the examples are plotted is sorted by the examples' learning speed. Faster-learned examples from the first task are less likely to be forgotten at the end of the second task. (d) The mean learning speed of examples in the first task vs. the % of networks that remember them at the end of the second task. Networks forget more examples learned slowly.

**Computing learning speed.** To calculate the learning speed of each example, we maintain a boolean matrix, called the epoch-wise classification matrix, $M \in \{0,1\}^{E \times |\mathcal{D}_t|}$ throughout the training on task $\mathcal{D}_t$, indicating whether the model correctly classified each example during learning. Specifically, for example $(x_i, y_i) \in \mathcal{D}_t$, we have $M_{e,i} = \mathbb{1}[f_t^e(x_i) = y_i]$. An example's learning speed is the average of the $M$ across epochs. This matrix can be computed during the forward pass of the network, incurring minimal computation overhead. Furthermore, memory usage can be reduced by directly computing the matrix's mean during training, maintaining a single float for each example.

## 2.2 PRELIMINARIES

**Class Incremental vs. Task Incremental Learning.** Although similar, task incremental learning differs from class incremental learning by providing the task identity for each example during testing, allowing the use of the specific classifier for the classes within the task. We conducted experiments for both class and task incremental scenarios. Since the qualitative results are similar and the observations remain consistent, we present the task incremental results in the main paper and include the class incremental results in App. A to avoid redundancy.

**Datasets.** We investigated various image continual learning classification tasks using split versions of several image datasets, including CIFAR-10, CIFAR-100 (Krizhevsky et al., 2009), and TinyImageNet (Le & Yang, 2015). The data is split into $T$ tasks by partitioning the classes into $T$ equal-sized subsets. This partitioning is denoted as dataset-T. For example, splitting CIFAR-10 into 5 classes is denoted as CIFAR-10-5, comprising 5 tasks, each with 2 distinct classes. Unless otherwise specified, classes are divided into tasks according to the original order they appeared in the dataset (often alphabetically).

**Architectures and hyper-parameters.** In our experiments, unless stated otherwise, we trained ResNet-18 for $E = 100$ epochs per task. We employed a base learning rate of 0.1 with a cosine scheduler, SGD optimizer, momentum of 0.9, and weight decay of 0.0005. All networks were trained on NVIDIA TITAN X. These hyperparameters were chosen arbitrarily based on their performance during joint dataset training and were consistent across all experiments. We anticipate consistent qualitative results despite potential variations in these hyperparameters. When introducing a replay buffer, we employed an experience replay strategy (Rolnick et al., 2019), alternating batches of data from the new task and the replay buffer. When integrating with other continual learning algorithms, we evaluated all methods within the framework of (Boschini et al., 2022; Buzzega et al., 2020), using the hyperparameters suggested in the original papers, changing only the replay buffer sampling to Goldilocks, keeping the rest of the method intact.

## 2.3 CORRELATING LEARNING SPEED WITH CATASTROPHIC FORGETTING

We begin by exploring catastrophic forgetting in a simplistic case, without using replay buffer ($B = \emptyset$), and with only $T = 2$ tasks. In Fig. 2a, we plot the mean test accuracy of each task when

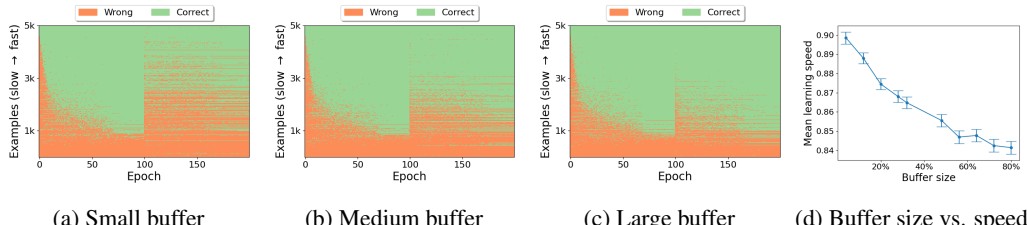

(a) Small buffer       (b) Medium buffer       (c) Large buffer       (d) Buffer size vs. speed

Figure 3: Impact of the buffer size on forgetting dynamics. (a-c) Similarly to Fig. 2c, the first task's binary matrices $M$, when adding replay buffer of varying buffer sizes: small ($1k$), medium ($3k$), and large ($12k$). The y-axes denote examples, and the x-axes denote epochs, indicating if the example has been classified correctly by the model at that epoch. The order in which examples appear is sorted by the examples' learning speed. In all cases, examples that were learned faster are more likely to be remembered. With bigger buffers, the networks can remember gradually slower-to-learn examples. (d) The mean learning speed of remembered examples of models with different buffer sizes. Models with bigger replay buffers remember slower-to-learn examples.

training 10 networks on CIFAR-100-2. The dashed black line marks the transition from the first to the second task. Consistent with prior research, the networks exhibit significant catastrophic forgetting, evidenced by a sharp decline in the accuracy of the first task during training on the second task. Our objective is to characterize those examples where the network successfully classified after the first task but failed after the second task.

To simultaneously track learning speed and forgetting, we maintain the epoch-wise boolean classification matrix $M$ for each network (formally defined in §2.2). Throughout the training of both tasks, we record, for each epoch, whether the network correctly classified each example from the first task. The initial 100 epochs represent classification during the first task, enabling us to observe the learning speed of the different examples, while the subsequent 100 epochs allow us to monitor the forgetting of the first task during the training of the second task.

In Figs. 2(b-c), we examine the epoch-wise classification matrices for the train and test sets, respectively. To aid visualization, instead of plotting the epoch-wise classification matrix for each network, we aggregate the networks from all runs using majority voting. The examples in the matrix are sorted by their learning speed. We observe a correlation between learning speed and catastrophic forgetting: examples learned more quickly during the first task tend to remain correctly classified throughout the second task, whereas slower-learned examples are prone to immediate misclassification upon task transition, making them susceptible to forgetting. Essentially, slower learning speeds of examples correlate with a higher probability of forgetting by deep models. Additional classification matrices, for different datasets and different amounts of tasks, can be found in App. B and Fig. 19.

To quantify this relationship, we define an example in the first task as "remembered" by the network if it is classified correctly at the conclusion of both the first and second tasks. In Fig. 2d, we plot, for each example in the test set, the % of networks that remembered it vs. its mean learning speed across all 10 networks. For visualization, we group examples remembered by a similar percentage of networks and average their mean learning speeds. We observe a strong correlation ($r = 0.995, p \leq 10^{-10}$), indicating that networks are more likely to remember quickly learned examples.

While the results presented here pertain to a simplified case, this phenomenon is robust across diverse datasets, architectures, and hyperparameters. For additional experiments, please refer to App. B, C.

## 2.4   HOW THE REPLAY BUFFER CHANGES THE CATASTROPHIC FORGETTING DYNAMICS

To evaluate replay-based methods, we incorporate a replay buffer in our experiments. Similarly to §2.3, we trained 10 networks on CIFAR-100-2, with replay buffers of varying sizes ($1k$, $3k$, $10k$). In Figs. 3(a-c), we plot the epoch-wise classification matrix of the test data for each case. The examples appear in each matrix in the order of their learning speed, from slow (bottom) to quick (top). As expected, integrating a replay buffer mitigates some catastrophic forgetting, boosting the model performance on the first task, with larger buffers yielding better results. Nevertheless, similar to the scenario without a replay buffer $B = \emptyset$, a relationship exists between learning speed and catastrophic forgetting: networks tend to remember fast-learned examples while forgetting those learned later even

---

**Algorithm 1** Training continual learning method $CL$ with Goldilocks sampling.

---

**Input:** task $\mathcal{D}_t$, sample size $|B|$, epochs $E$, amount of quick/slow learned examples to remove $q, s$.
**Output:** buffer of size $|B|$

1: $Classification\_Matrix \leftarrow 0^{E \times |\mathcal{D}_t|}$
2: **for** $e = 1, ..., E$ **do**
3:     Train the model $f$ one epoch using the selected continual learning method $CL$
4:     **for** $(x_i, y_i) \in \mathcal{D}_t$ **do**                $\triangleright$ Can be calculated efficiently in the forward pass
5:         Classification_Matrix$[e, i] \leftarrow \mathbb{1}[f(x_i) = y_i]$
6:     **end for**
7: **end for**
8: $Learning\_Speed \leftarrow \text{Mean}(Classification\_Matrix, \text{axis=0})$
9: $\mathcal{D}_t^{'} \leftarrow$ remove $q\%$ highest and $s\%$ lowest $Learning\_Speed$ examples from $\mathcal{D}_t$
10: **return** $|B|$ examples sampled uniformly from $\mathcal{D}_t^{'}$

---

when using a replay buffer. Notably, models with larger buffer sizes remember most of the examples that models with smaller buffer sizes do, while additionally remembering gradually slower-learned examples. Similar results occur when changing the network architecture (see App. B, C).

We quantify the relationship between the buffer size and remembered examples' learning speed. Training 10 networks on CIFAR-100-2 with various buffer sizes, we plot the mean learning speed of examples remembered by the models for each buffer size. A robust correlation emerges ($r = 0.966$, $p \leq 10^{-6}$): models with smaller buffer sizes remember quickly learned examples, whereas larger models with larger buffers enable the model also to remember slower-learned. These results are plotted in Fig. 3d. Similar results were achieved across diverse datasets, architectures, hyperparameters, and continual learning algorithms, see App. B, C.

## 3 EFFECTS OF BUFFER COMPOSITIONS ON CATASTROPHIC FORGETTING

Replay-based continual learning methods typically sample the replay buffer uniformly, including both quickly and slowly learned examples. In this section, we analyze various replay buffer compositions and find that medium-speed learned examples are the most beneficial for the replay process. These examples, which the model has only a moderate chance of remembering, prove particularly important. One possible explanation is that increasing the buffer size impacts the catastrophic forgetting of these examples the most, suggesting that focusing on them is advantageous.

### 3.1 THE GOLDILOCKS APPROACH – METHODOLOGY AND DEFINITIONS

To explore different buffer compositions, we remove a combination of the quickest and slowest learned examples from the data and then sample the buffer uniformly from the remaining examples. This allows us to add a bias to the replay buffer toward examples learned at certain speeds. We propose Goldilocks, a novel sampling method, used both in our empirical analysis and as a practical replay buffer sampling technique.

Formally, Goldilocks is defined by two hyperparameters, *quick* and *slow*, which specify the percentage of examples learned too quickly or too slowly to be removed from dataset $\mathcal{D}_t$. During the training of the model $f$ on $\mathcal{D}_t$, we maintain the learning speed of each example. At the end of task $t$, examples in $\mathcal{D}_t$ are sorted by learning speed, and the top $\%quick$ and bottom $\%slow$ are removed. The pseudocode for Goldilocks sampling is provided in Algorithm 1.

With Goldilocks, the *quick* and *slow* hyperparameters are fixed. However, by keeping a constant buffer size and varying these hyperparameters across experiments, we create replay buffers with different compositions, focused on either slowly or quickly learned examples. Next, we explore a range of buffer compositions for each method, analyzing their ability to mitigate forgetting.

### 3.2 WHICH EXAMPLES SHOULD APPEAR IN THE REPLAY BUFFER

We compare replay buffer compositions for different buffer sizes using the CIFAR-100-2 dataset. In Figs. 4(a-b), we plot the mean final test accuracy of 10 networks for each buffer composition. The

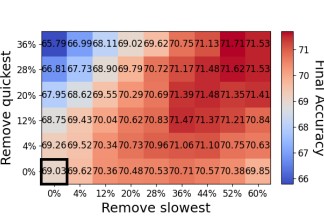

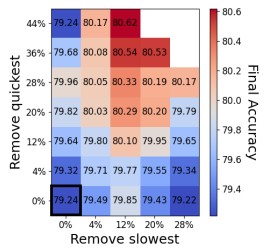

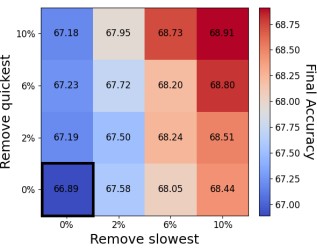

(a) CIFAR-100-2, small buffer     (b) CIFAR-100-2, big buffer     (c) TinyImageNet-2, big buffer

Figure 4: Comparison of buffer compositions for various buffer sizes, removing slowly and quickly learned examples (see §3.1). The mean final accuracy of 10 networks is plotted. The baseline (uniformly sampled buffer) appears at $(0\%, 0\%)$, marked with a black box. (a) CIFAR-100-2 with a small $1k$ buffer: removing slower-to-learn examples is more beneficial. (b) CIFAR-100-2 with a large $10k$ buffer: removing faster-to-learn examples is more beneficial. (c) TinyImageNet-2 with a $20k$ buffer. Across all scenarios, diverse buffer compositions are beneficial, with smaller buffers benefiting more from prioritizing faster-to-learn examples.

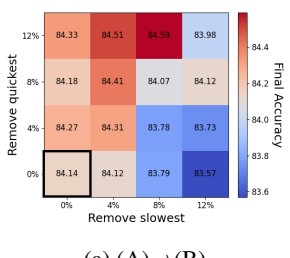

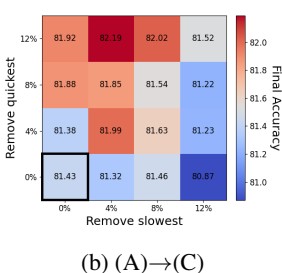

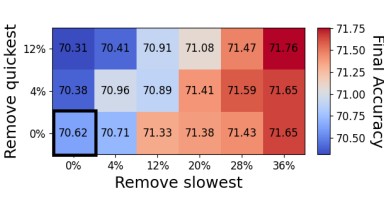

(a) (A)→(B)     (b) (A)→(C)     (a) GEM, CIFAR-100-2, 500 buffer

Figure 5: Comparison of buffer compositions when training on the same initial task, followed by different subsequent tasks. We use three subsets of CIFAR-100: A, B, and C (see §3.3). (a) Task A followed by B. (b) A followed by C. Despite the differences in subsequent tasks, the buffer compositions for the original task similarly mitigate catastrophic forgetting.

Figure 6: Comparison of buffer compositions for GEM with a 500-size buffer. Similar to Fig. 4, a wide range of compositions is beneficial. With the smaller buffer, removing mostly slower examples yields better results.

point $(slow, fast) = (0, 0)$ (marked by a black box) represents training with a uniformly sampled replay buffer. To aid visualization, standard errors are plotted separately (see App. D). In both cases, a wide range of buffer compositions significantly improves the final accuracy, suggesting that uniform sampling of the replay buffer is suboptimal.

While many buffer compositions are beneficial for both cases, we see that when the buffer size is smaller, it is more beneficial to remove slowly-learned examples, focusing mainly on the easier examples learned quickly by the models. In contrast, with larger buffers, the optimal focus in the buffer shifts towards harder, slowly-learned examples, where quickly-learned examples are removed.

In Fig. 4c, we replicate this experiment on TinyImageNet-2, with a large buffer of size $20k$. Similar to CIFAR-100-2, focusing on examples learned midway through learning yields the best performances. Notably, across all cases, the range of hyperparameters that enhance performance is broad and continuous, indicating that improvements are not limited to specific hyperparameter sets.

To assess the robustness of these results, we repeat the experiment from Fig. 4 under various learning settings, including changes to optimizers, architectures, learning rates, training durations, fine-tuning parameters, regularization techniques, and data splits. Across all cases, we observe consistent qualitative results. Details of these experiments are provided in App. C.

**Discussion.** Our findings suggest that a good composition of a replay buffer consists of examples learned during the middle of the training – neither too quickly nor too slowly. A possible intuition for this observation may be connected to simplicity bias. As networks train, they gradually acquire the ability to handle increasingly complex concepts. By the end of a task, the network performs well on examples up to a certain level of complexity. For replay buffers, this implies that examples

learned too quickly are overly simplistic, offering limited additional value since the network has already mastered more complex tasks. Conversely, examples learned too slowly are suboptimal, as the network struggled to learn them initially, so it is likely to struggle when replaying them.

## 3.3 INDEPENDENCY OF THE OPTIMAL BUFFER COMPOSITION ON SUBSEQUENT TASKS

We empirically examine how subsequent tasks impact the ideal composition of a replay buffer for a given task. We find that regardless of the similarity or dissimilarity between subsequent tasks and the original task, the optimal replay buffer composition remains largely independent and consistent. To show this, we conduct experiments with three distinct tasks denoted A, B, and C. By training the model on task A and subsequently introducing either task B or task C while keeping the original task unchanged, we analyze the ideal replay buffer composition under varying subsequent tasks.

We initially focus on tasks from the same datasets. We pick task A to be the first 25 classes of CIFAR-100, task B to be classes 26 to 50, and task C to be classes 51 to 75. In Fig. 5, we depict the mean final accuracy of 10 networks across various replay buffer compositions when using either task B or task C as the subsequent task after task A. Despite some quantitative variations in performance, the qualitative results consistently indicate that good buffer compositions for one case remain effective for the other, and vice versa. These results suggest that the ideal replay buffer composition of task A is robust to differences between tasks B and C in this context.

Further, we explore scenarios involving either changing the classification task or replacing it with memorization by training on random labels. We adopt the RotNet approach (Gidaris et al., 2018) to change the classification task. We pick tasks A and B to be different subsets of CIFAR-100, while task C involves a rotation classification task, where each example in task A is randomly rotated between $90°, 180°$, or $270°$ degrees, with the task being to classify these rotations regardless of the original label. For the random case, tasks A and B remain unchanged, while task C involves the same subset of task B, but with labels chosen uniformly at random, as suggested in (Zhang et al., 2021). In both scenarios, we observe consistent results: the ideal replay buffer composition of task A remains robust to differences between tasks B and C in both contexts. Further details and heatmaps for the results are available in App. E.

**Choosing hyperparameters.** The broad range of $quick$ and $slow$ values that yield effective buffer compositions allows for consistent hyperparameter settings across datasets and tasks, minimizing the need for fine-tuning. For instance, selecting $quick = slow = 20\%$ – among other viable options – consistently enhances performance across all evaluated datasets and scenarios and generalizes well to unseen datasets. These values can also be further adjusted heuristically based on prior data knowledge: for larger buffers and simpler datasets, prioritizing the removal of slowly-learned examples is advantageous, while for smaller buffers or more complex datasets, focusing on quickly-learned examples proves more effective. If computational resources allow, systematic tuning of these hyperparameters is feasible. By creating a rotated version of the dataset, we can simulate a 2-task scenario using the original and rotated data, then perform a grid search to fine-tune the hyperparameters. The smooth effect of these parameters on performance suggests that a coarse grid search is usually sufficient.

## 3.4 EXTENDING GOLDILOCKS TO SAMPLING TO MULTI-TASK SCENARIOS

In many continual learning problems, models encounter long sequences of tasks. To extend Goldilocks sampling to multitask scenarios, we measure the learning speed of each example only during the training of the task it belongs to. In theory, this allows for separate $quick$ and $slow$ hyperparameters for each task, determining how many examples to remove when sampling for the current buffer. However, tuning multiple hyperparameters for each task is computationally expensive. Thus, in practice, we apply the same $quick$ and $slow$ hyperparameters across all tasks when dealing with more than two tasks.

In multitask training, we also need an effective method to remove examples from the buffer, as the proportion of examples from each task diminishes as the number of tasks increases. In Goldilocks, examples can be removed uniformly at random, while still adhering to the principle that neither too-fast nor too-slow examples remain in the buffer.

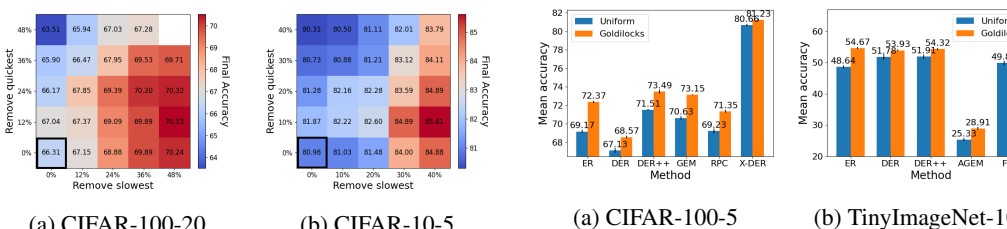

(a) CIFAR-100-20     (b) CIFAR-10-5       (a) CIFAR-100-5     (b) TinyImageNet-10

Figure 7: Comparison of buffer compositions in multi-task training scenarios: (a) CIFAR-100-20, (b) CIFAR-10-5, both trained with a 5k buffer. Similarly to the 2-task case, a wide range of buffer compositions significantly improves performance, helping to mitigate catastrophic forgetting. The performance gains of Goldilocks tend to get bigger in the multi-task case.

Figure 8: Goldilocks vs. uniform sampling with various continual learning methods. Bars show the mean final test accuracy across all tasks, with random sampling in blue, and Goldilocks in orange. Error bars denote the standard error. (a) CIFAR-100-5 (b) TinyImageNet-10, with a buffer of 500 examples. Goldilocks improves performance across all methods and datasets.

Fig. 7 presents the final test accuracy for different buffer compositions on CIFAR-100-20 and CIFAR-10-5. In both cases, we observe trends similar to the two-task scenario – many different buffer compositions significantly improve final accuracy, helping mitigate catastrophic forgetting. Interestingly, the performance boost is even more pronounced in the multi-task scenario, suggesting that Goldilocks is particularly effective in such settings.

## 3.5 GOLDILOCKS SAMPLING IMPROVES VARIOUS CONTINUAL LEARNING METHODS

Many replay-based continual learning algorithms sample the replay buffer uniformly. In this section, we investigate the impact of replacing uniform sampling with Goldilocks in various such algorithms, including DER, DER++ (Buzzega et al., 2020), X-DER (Boschini et al., 2022), AGEM (Chaudhry et al., 2019), ER (Rolnick et al., 2019), GEM (Lopez-Paz & Ranzato, 2017), RPC (Pernici et al., 2021), and FDR (Titsias et al., 2020). Our results demonstrate that Goldilocks consistently improves the performance of all tested continual algorithms, thereby advancing the state of the art in multiple image classification tasks.

We replicate the experimental setup outlined in §3.2, exploring various compositions of the replay buffer in a grid-like fashion. In Fig. 6 we plot a heatmap example for GEM trained on CIFAR-100-2, employing a buffer size of 500. We focus on small buffers, as the original works often mainly report results for such buffers. Our findings mirror our previous findings on the experience replay case: a large set of hyperparameters improves the algorithm's performance, while prioritizing the removal of slowly learned examples proves advantageous given the buffer's small size. Comparable outcomes across different algorithms, datasets, and class incremental learning scenarios are detailed in Apps. A,F.

Next, we evaluate continual learning methods combined with Goldilocks on standard benchmarks, including CIFAR-10-5, CIFAR-100-5, and TinyImageNet-10, using a small buffer size of 500, as done in the original works. For buffer management, we applied either random sampling or reservoir sampling, consistent with what was done in the original code of each method. The *quick* and *slow* hyperparameters were determined through RotNet training, as described in §3.3. Similarly to the other multi-task cases, we kept the *quick* and *slow* hyper-parameters constant across tasks. Goldilocks consistently enhances the performance of various continual learning algorithms, demonstrating its compatibility with a range of methods. These results are shown in Fig. 8.

## 3.6 COMPARING GOLDILOCKS TO OTHER REPLAY BUFFER SAMPLING METHODS

While numerous replay buffer sampling methods have been proposed over the years, uniform sampling remains prevalent in practical implementation within continual learning algorithms. This preference may stem from the specific settings or algorithmic modifications required for other sampling methods to work well. These methods do not consistently improve different continual learning algorithms

Table 1: Comparison of replay buffer sampling methods. Each entry represents the mean test accuracy across all tasks for 10 networks trained with experience replay using different sampling methods. The highest accuracy in each scenario is in bold. For visualization, standard errors are reported separately in Table 2. While certain methods, such as Herding (small buffers) and GSS (large buffers), perform well in specific cases, they fail to consistently outperform random sampling across all scenarios. In contrast, Goldilocks consistently achieves the highest accuracy across all scenarios.

| | CIFAR-10-2 | | CIFAR-100-2 | | CIFAR-100-20 | | CIFAR-10-5 | | TinyImageNet-2 | |
|---|---|---|---|---|---|---|---|---|---|---|
| Buffer size | $1k$ | $10k$ | $1k$ | $10k$ | $1k$ | $10k$ | $1k$ | $10k$ | $1k$ | $10k$ |
| Random | 87.67 | 95.03 | 69.03 | 79.24 | 51.75 | 71.25 | 79.48 | 82.4 | 49.81 | 61.48 |
| Max entropy | 84.44 | 94.63 | 64.27 | 77.91 | 48.3 | 69.83 | 77.28 | 81.72 | 44.49 | 57.88 |
| IPM | 85.9 | 94.76 | 64.16 | 75.73 | 49.91 | 71.7 | 79.3 | 80.57 | 48.58 | 61.97 |
| GSS | 85.28 | 95.55 | 64.15 | 80.11 | 48.13 | 72.9 | 76.71 | 84.13 | 49.36 | 62.89 |
| Herding | 88.41 | 94.53 | 71.07 | 78.97 | 54.0 | 73.51 | 81.8 | 81.03 | 51.17 | 62.44 |
| Goldilocks | **88.99** | **96.03** | **71.62** | **80.62** | **55.43** | **74.80** | **83.59** | **89.17** | **52.15** | **63.16** |

across datasets and buffer sizes. In contrast, Goldilocks sampling improves a wide range of continual learning algorithms across various buffer sizes and datasets.

We explored various replay buffer sampling strategies, including uniform sampling, max entropy, IPM (Zaeemzadeh et al., 2019), GSS (Aljundi et al., 2019), and Herding (Rebuffi et al., 2017). We trained 10 networks on CIFAR-10-2, CIFAR-100-2, TinyImageNet-2, CIFAR-100-20, and CIFAR-10-5 datasets with buffer sizes of $1k$ and $10k$ using experience replay. Table 1 presents the mean test accuracy across all tasks for each case.

Goldilocks consistently outperforms all other sampling strategies across all scenarios, surpassing them in all four cases. In contrast, other sampling methods exhibit efficacy only in specific scenarios. For instance, Herding, which prioritizes examples with mean features close to class mean features, performs better than uniform sampling with smaller buffer sizes but worse with larger buffers. This might be because this preference for typical examples, which are learned early on, is more beneficial with smaller buffers. Conversely, GSS, which selects examples based on gradient distances, outperforms uniform sampling in scenarios with large buffers but performs suboptimally with smaller budgets. This bias towards selecting examples with large gradients favors examples learned slowly, which is more beneficial with large buffers.

## 4 DISCUSSION

We show a correlation between the speed at which examples are learned and the likelihood of models forgetting these examples when learning new data. This correlation aligns closely with the simplicity bias, suggesting that models tend to forget more complex examples. Although measuring *learning speed* is a simple proxy for example complexity, its simplicity underscores the potential for developing more sophisticated methods to mitigate catastrophic forgetting, offering avenues for future research.

While most replay-based continual learning works focus on better utilization of the replay buffer, our study demonstrates the potential for buffer improvement, extending the scope of continual learning research beyond buffer utilization alone. By showing that Goldilocks can be implemented with minimal computational overhead, we highlight its applicability to future advancements in continual learning. Furthermore, this work underscores the mutual benefits of concurrent progress in better sampling of the replay buffer and improvements in buffer utilization, suggesting promising directions for future research.

**Limitations.** One limitation of Goldilocks is that the *learning speed* score is based on the number of training epochs. With a small number of epochs, the score becomes overly discrete, which reduces the precision of the evaluation. However, as shown in App. C and Fig. 22, Goldilocks remains effective even with a small number of epochs, as long as the network converges, although with increased noise. Nonetheless, many continual learning scenarios emphasize stream-like settings, which typically involve only a single epoch. These scenarios are not well-supported by Goldilocks, as the *learning speed* fails to adequately capture how different examples are learned over time. Alternative methods for approximating example complexity are needed, which we leave as future work.

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

APPENDIX

## A  CLASS INCREMENTAL LEARNING

There are two primary paradigms in continual learning: class incremental learning (CIL) and task incremental learning (TIL). The distinction lies in how tasks are handled. In TIL, task identities are known during inference, enabling the use of specialized classifiers within each task, whereas in CIL, the model must infer both the task and class identity without prior knowledge, often leading to lower performance. These paradigms address different real-world scenarios, and methods optimized for one sometimes underperform on the other, necessitating separate evaluations in continual learning research.

Our work examines the relationship between the speed at which examples are learned and their susceptibility to forgetting in continual learning. Despite the differences between CIL and TIL, our study reveals that the same qualitative results and the same conclusions can be made in both cases. Therefore, to avoid repetition in the main text, we focus there on TIL only, as it isolates forgetting

at the model level rather than conflating it with final-layer interference. To complement this, we reproduce all key experiments under CIL settings in this appendix, highlighting similar conclusions despite the inherent challenges of CIL.

Figs. 9, 10, 11, 12, 13 present CIL counterparts to the main text's results, Figs. 2, 3, 4, 6, 7 respectively. While CIL results exhibit lower overall performance, the trends remain consistent: examples learned faster are less likely to be forgotten, and the correlations between the buffer size and forgetting and the correlation between the learning speed and forgetting are strong. Notably, optimal buffer composition in CIL scenarios (Figs. 11, 12, 13) slightly favors keeping more quickly learned examples compared to TIL, reflecting the added complexity of task inference in CIL.

Finally, Fig. 14 reproduces results from Fig. 8, demonstrating the impact of our Goldilocks sampling strategy across CIL datasets and algorithms. Goldilocks consistently yields significant performance gains, reaffirming its effectiveness across diverse continual learning challenges.

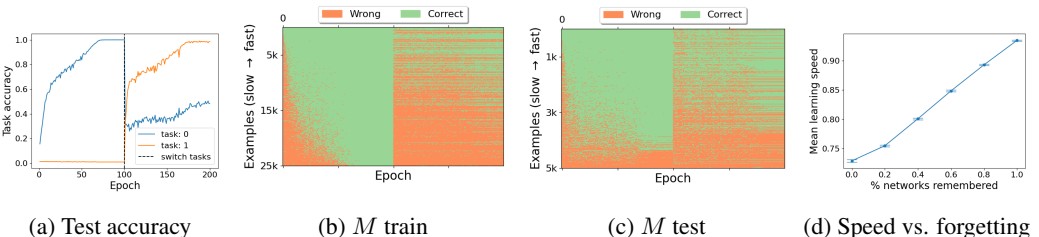

(a) Test accuracy    (b) $M$ train    (c) $M$ test    (d) Speed vs. forgetting

Figure 9: Repeating Fig. 2 in class incremental settings. All results are trained on CIFAR-100-2, without a replay buffer. (a) Mean test accuracy of each task. (b-c) first task's binary epoch-wise classification matrices $M$ for train and test data. (d) The mean learning speed of examples in the first task vs. the % of networks that remember them at the end of the second task.

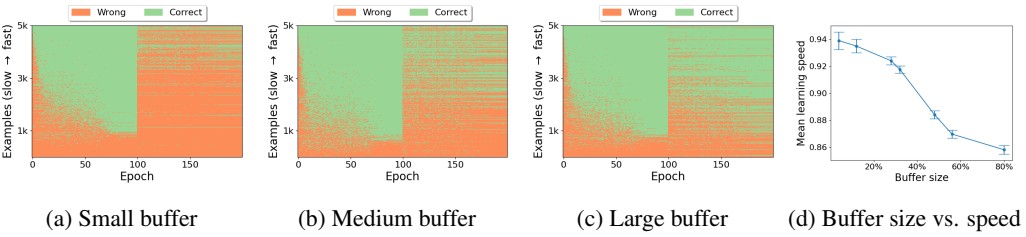

(a) Small buffer    (b) Medium buffer    (c) Large buffer    (d) Buffer size vs. speed

Figure 10: Repeating Fig. 3 for class incremental learning. Impact of the buffer size on forgetting dynamics. (a-c) The first task's binary matrices $M$, when adding replay buffers of varying buffer sizes. (d) The mean learning speed of remembered examples of models with different buffer sizes. Models with bigger replay buffers remember slower-to-learn examples.

## B  CORRELATIONS BETWEEN LEARNING SPEED AND CATASTROPHIC FORGETTING IN DIFFERENT DATASETS

In Section 2, we established a link between the *learning speed* of examples and catastrophic forgetting, primarily utilizing ResNet-18 networks trained on CIFAR-100-2. Here, we expand upon these findings by demonstrating similar phenomena across different datasets, architectures, and task numbers.

**Other Datasets.** We first explore other datasets using the experimental setup detailed in Fig. 2. We extend our analysis to CIFAR-10-2, CIFAR-100-2, and a subset of TinyImageNet-2 (comprising the initial 40 classes). The extended epoch-wise classification matrices for each task and the correlation between *learning speed* and the percentage of networks remembering each example are plotted in Fig. 15. Consistently across all three datasets, we observe a robust correlation between *learning speed* and the likelihood of example retention during continual training. Notably, faster-learned examples exhibit lower rates of catastrophic forgetting, as evidenced by both quantitative correlations and visual inspection of the extended epoch-wise classification matrices.

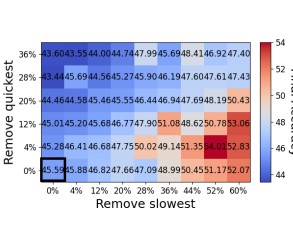 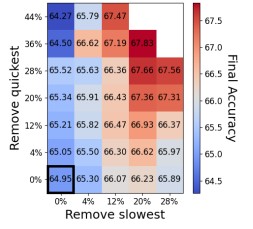

(a) CIFAR-100-2, small buffer     (b) CIFAR-100-2, big buffer

Figure 11: Repeating Fig. 4 for class incremental learning. Comparison of buffer compositions for various buffer sizes. (a) CIFAR-100-2 with a small $1k$ buffer. (b) CIFAR-100-2 with a large $10k$ buffer. Note that since class incremental learning is harder than task incremental learning, it is beneficial to remove fewer "quick" examples in this case.

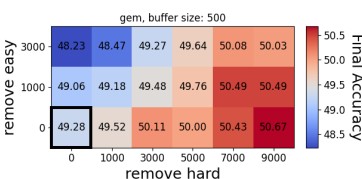

(a) GEM, CIFAR-100-2, 500 buffer

Figure 12: Repeating Fig. 6 for class incremental learning. The results mirror the task-incremental case, although with lower performances.

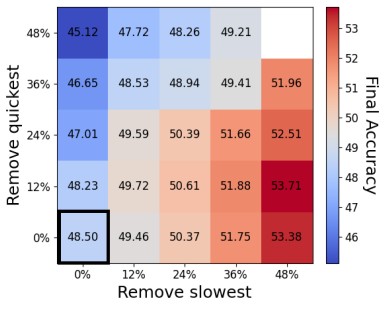 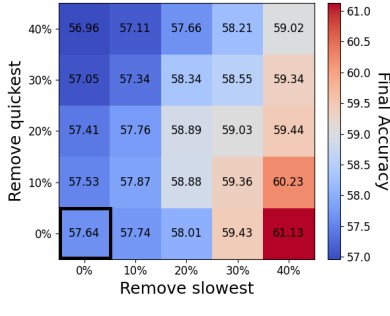

(a) CIFAR-100-20        (b) CIFAR-10-5

Figure 13: Repeating Fig. 7 for class incremental learning. (a) CIFAR-100-20, (b) CIFAR-10-5, both with a 5k buffer. Similar to the 2-task case, a wide range of buffer compositions significantly improves performance, helping to mitigate catastrophic forgetting.

Further, in Fig. 16, we expand upon Fig. 3 to include CIFAR-10-2. This extension reveals that as the buffer size increases, continual models tend to retain slower-to-learn examples. Analogous to the results observed in CIFAR-100-2, as depicted in Fig. 3, we also observe a near-perfect negative correlation between the buffer size and the mean learning speed of remembered examples in CIFAR-10-2.

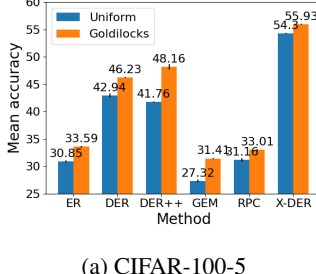 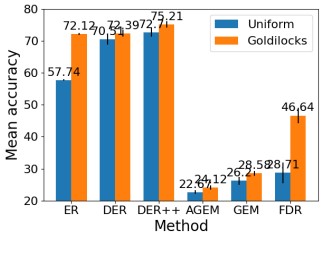 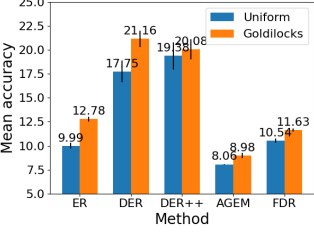

(a) CIFAR-100-5      (b) CIFAR-10-5      (c) TinyImageNet-10

Figure 14: Goldilocks vs. uniform sampling with various continual learning methods, in class-incremental settings. Each bar shows the mean final test accuracy across all tasks for each method, with a uniform sampled buffer (traditionally done) in blue, and Goldilocks in orange. Error bars denote the standard error.

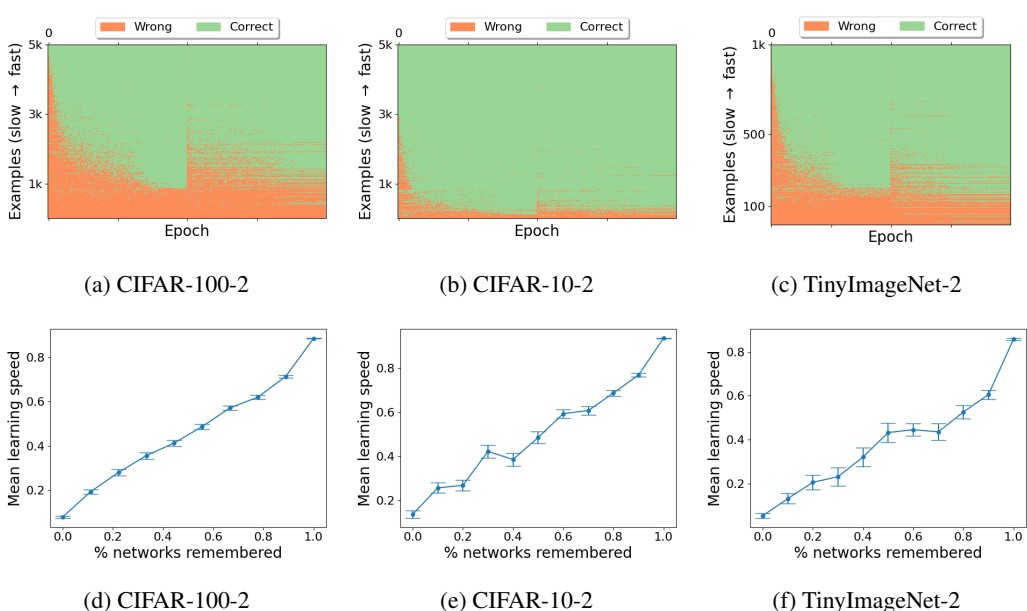

Figure 15: Extendeing Fig. 2 to CIFAR-100-2, Cifar-10-2 and a subset of TinyImageNet-2. (top) The first task's binary epoch-wise classification matrices $M$ for the test data of each dataset. The y-axis denotes examples, and the x-axis denotes epochs, indicating if an example is correctly classified by the model at the given epoch. The order in which examples appear is sorted by *learning speed*. Faster-learned examples from the first task are less likely to be forgotten at the end of the second task. (bottom) The mean *learning speed* of examples in the first task vs. the percentage of networks that remember them at the end of the second task. Networks forget more examples learned slowly across all 3 datasets.

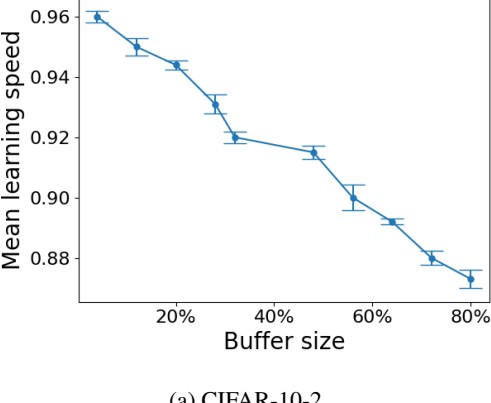

(a) CIFAR-10-2

Figure 16: Extending Fig. 3d to CIFAR-10-2. We plot the mean *learning speed* of remembered examples by 10 models trained with different buffer sizes. Models with bigger replay buffers remember slower-to-learn examples.

**Classification matrices of different datasets.** In Section 2, we analyze epoch-wise classification matrices, which indicate whether each example was classified correctly or incorrectly at each epoch during training. Figs. 2 and 3 plot these matrices for CIFAR-100-2 and CIFAR-10-2, revealing a clear correlation between learning speed and catastrophic forgetting: examples learned quickly are less likely to be forgotten during task switches.

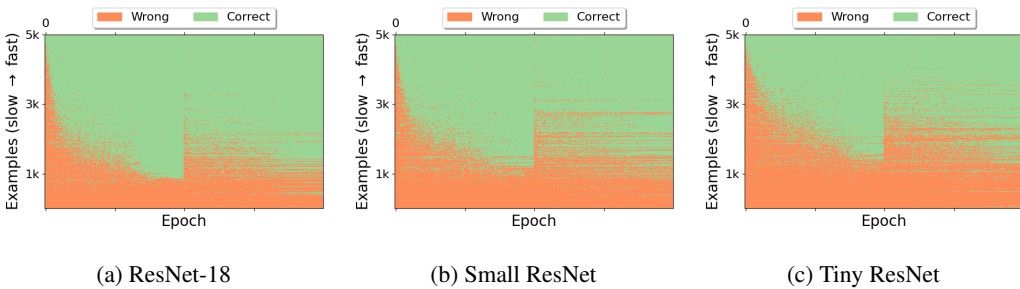

(a) ResNet-18          (b) Small ResNet          (c) Tiny ResNet

Figure 17: Comparing the extended epoch-wise classification matrix for different architectures, the y-axis represents examples, and the x-axis represents epochs, indicating if an example is correctly classified by the model in a given epoch. In (a), we train ResNet-18. In (b), we train ResNet-18 with both width and depth reduced by half. In (c), we train ResNet-18 with both width and depth reduced by a factor of four. In all cases, the order examples appear is sorted by *learning speed*. Examples learned quickly in the first task are less likely to be forgotten after the second task.

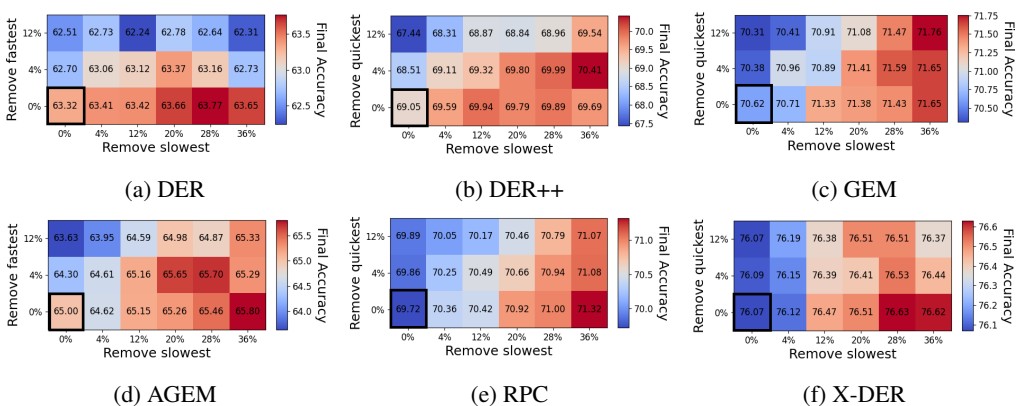

(a) DER          (b) DER++          (c) GEM

(d) AGEM          (e) RPC          (f) X-DER

Figure 18: Comparing different replay buffer compositions for various continual learning methods, with a buffer of size $500$. The experimental setup replicates the one done in Fig. 4. Like the experience replay case, focusing on examples learned midway through the learning process is most beneficial. Due to the smaller buffer, removing slower examples is better.

In Fig. 19, we extend this analysis to the classification matrices of the first task of TinyImageNet-2, CIFAR-100-20, and CIFAR-10-5. For these datasets, which involve more than two tasks, the x-axis includes multiple task switches, marked by vertical dashed black lines. The same correlation is observed and becomes even more pronounced: examples from the first task are increasingly forgotten as additional tasks are introduced. With each new task, the newly forgotten examples are increasingly those learned earlier in training. Notably, examples learned fastest remain robust to forgetting, even after a long sequence of tasks, while those learned more slowly are more vulnerable.

**Other Architectures.** We extend the results from Fig. 2 to architectures beyond ResNet-18 by creating two new, smaller variants: Small-ResNet and Tiny-ResNet, which can be found in Fig. 17. These are formed by reducing both the width and depth of ResNet-18 by a factor of 2 and 4 respectively. We replicate the experimental setup from Fig. 2, training 10 models of each architecture on CIFAR-100-2 without a replay buffer, and storing the extended epoch-wise classification matrix. The order examples appear in the matrix is sorted by the *learning speed* of the examples. All matrices depict the classification of the test dataset. In all architectures, *learning speed* is highly correlated with catastrophic forgetting: networks forget more examples learned later in training while retaining an almost perfect recollection of those learned early on. Additionally, stronger architectures, similar to larger buffers, can remember slower-to-learn examples. This reinforces the motivation for Goldilocks, as it shows that the examples selected by Goldilocks are those the model could learn independently if it had a stronger architecture.

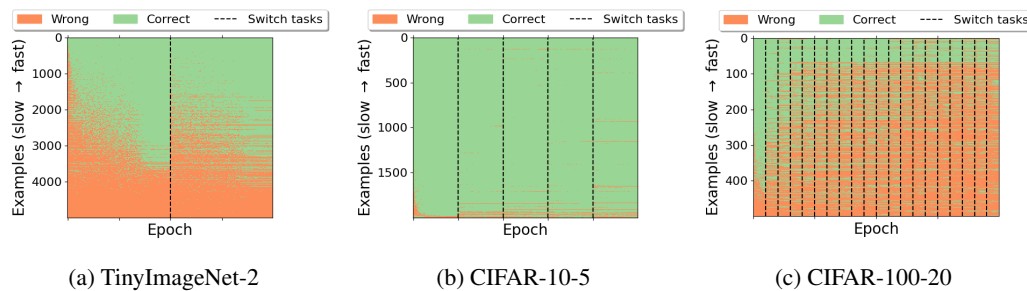

(a) TinyImageNet-2      (b) CIFAR-10-5      (c) CIFAR-100-20

Figure 19: Repeating Fig. 2c with different datasets and task counts: (a) TinyImageNet with 2 tasks, (b) CIFAR-10 with 5 tasks, and (c) CIFAR-100 with 20 tasks. Dashed black lines indicate the introduction of a new task. Consistent with Fig. 2c, examples learned more slowly are more prone to forgetting after task transitions. As the number of tasks increases, even examples learned relatively quickly become susceptible to forgetting.

**Other continual learning algorithms.** In §3.5, we demonstrated the benefits of non-uniform buffer compositions for various continual learning methods. We extend these findings, initially shown for GEM in Fig. 6, to additional algorithms including DER, DER++, GEM, A-GEM, RPC, and X-DER. For each algorithm, we trained 10 networks with different buffer compositions by varying the *quick* and *slow* hyperparameters of Goldilocks. For each algorithm, we trained 10 networks on different buffer compositions, achieved by varying the *quick* and *slow* hyperparameters of Goldilocks. In all cases, we used the hyperparameters and architectures suggested in the original works. We used a buffer of size 500, due to its popularity in previous works. Consistent with the GEM results, we found that replay buffers focused on examples learned mid-way through the training process were most beneficial. Additionally, in all cases, removing slower-to-learn examples proved advantageous, given the small buffer size. These results can be found in Fig. 18.

**Different number of tasks.** In most experiments presented in the paper, we focused on continual learning with 2 tasks. When considering a larger number of tasks, additional hyperparameters emerge, as the *quick* and *slow* parameters apply to each intermediate task. These can be tuned similarly to the two-task case, either through grid-search with a rotation task or using simple heuristics. To avoid excessive hyperparameter tuning when training on more than two tasks, we tuned the *quick* and *slow* values for the first task and kept them constant for all subsequent tasks. Results from training on CIFAR-10-5, CIFAR-100-5, and TinyImageNet-10 are shown in Fig. 27 in App. F. We observe that Goldilocks improves over uniform sampling in these cases as well. Notably, further tuning of all Goldilocks hyperparameters could yield even better results.

## C  THE EFFECTS OF DIFFERENT LEARNING HYPER-PARAMETERS

In this section, we investigate how varying learning hyperparameters influences the optimal composition of examples in the replay buffer. We replicate the experiment shown in Fig. 4a, training networks on CIFAR-100-2 with a buffer size of 1000, while modifying factors such as network architecture, optimizer choice, learning rate, regularization strength, and the number of training epochs.

Across all configurations, we observe consistent qualitative trends similar to those in Fig. 4a: a wide range of buffer compositions significantly improves performance and alleviates catastrophic forgetting. These findings suggest that the conclusions drawn in the main paper are robust and generalizable, extending to continual learning scenarios under different hyperparameter settings.

**Optimizers.** In Fig. 20, we compare the performance of different buffer compositions when training ResNet-18 on CIFAR-100-2 using three optimizers: SGD, Adam, and Adagrad. For SGD, we used a momentum of 0.9, a weight decay of 0.0005, and a learning rate of 0.1, as is common for CIFAR-100 training. For Adagrad, we employed a learning rate of 0.01 and a weight decay of 0.0001. For Adam, we used a learning rate of 0.001, a weight decay of 0.0001, and $(0.9, 0.999)$ betas. These hyperparameters were tuned to optimize ResNet-18's performance on CIFAR-100 without

considering continual learning, as the optimal settings vary by optimizer due to their differing update mechanisms.

Our results indicate that similar buffer compositions consistently enhance performance across all three optimizers. This consistency suggests that the benefits of our analysis are robust to the choice of optimizer, further supporting the generalizability of our findings.

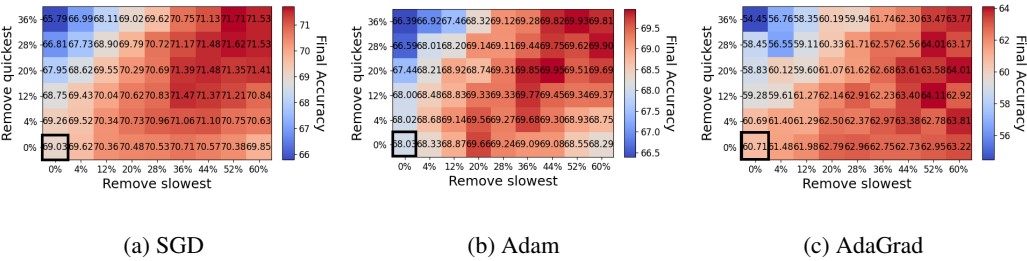

(a) SGD        (b) Adam        (c) AdaGrad

Figure 20: Repeating Fig. 4a using different optimizers: (a) SGD, (b) Adam, and (c) AdaGrad. Across all optimizers, a wide range of buffer compositions consistently enhances performance, yielding similar qualitative results to the experiment in the main text.

**Architectures.** In Fig. 21a, we evaluate the impact of buffer composition on the performance of three different architectures: ResNet-18, VGG-16, and a smaller version of ResNet, tiny-ResNet. All models were trained on CIFAR-100-2 under identical conditions. Our results show that similar buffer compositions consistently improve learning performance across these diverse architectures. This finding suggests that our analysis is not specific to any single architecture.

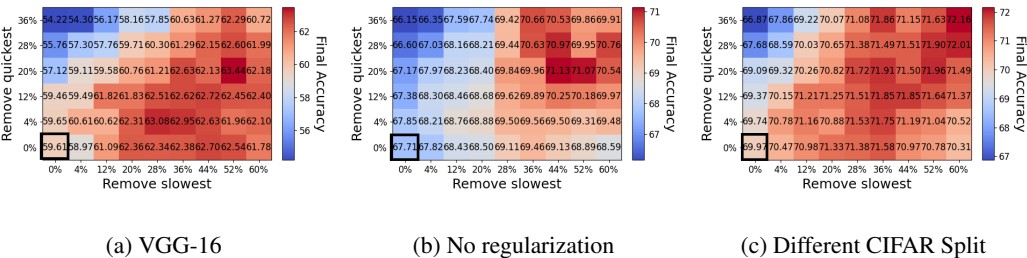

(a) VGG-16        (b) No regularization        (c) Different CIFAR Split

Figure 21: Repeating Fig. 4a under different learning settings: (a) replacing the ResNet-18 architecture with VGG-16, (b) removing weight decay from ResNet-18, and (c) using a random class-to-task split of CIFAR-100-2. In all cases, the results remain qualitatively similar, highlighting the robustness of the analysis.

**Training epochs.** In Fig. 22, we compare the performance of models trained for different numbers of epochs per task, all using a cosine learning rate scheduler to ensure exposure to both large and small learning rates throughout training. While reducing the number of epochs leads to a general drop in performance, we observe that similar buffer compositions consistently yield strong results, even under shorter training schedules. It is worth noting that the accuracy of determining the *learning speed* of each example may decrease with fewer epochs, as there is less time for the model to adapt. Nonetheless, the qualitative trends remain robust, with specific buffer compositions consistently outperforming or underperforming the random baseline across all tested epoch counts.

**Learning Rates.** We examine the impact of learning rates on model performance using SGD with rates of 0.05 and 0.2 (Figs. 23a, 23b). While the final accuracies differ across these configurations, the qualitative behavior remains consistent: the same buffer composition consistently achieves strong results.

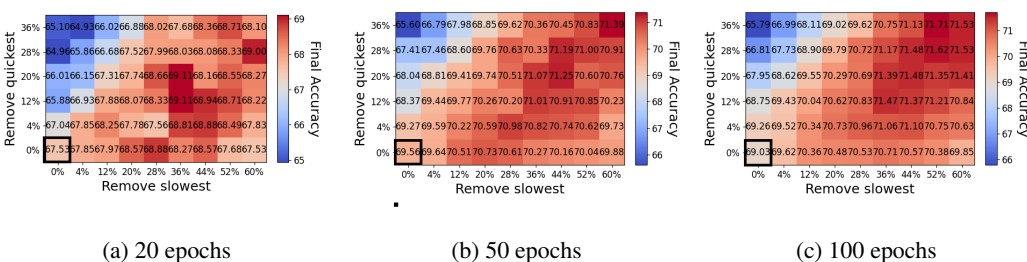

(a) 20 epochs      (b) 50 epochs      (c) 100 epochs

Figure 22: Repeating Fig. 4a with varying training epochs: (a) 20 epochs, (b) 50 epochs, and (c) 100 epochs per task. Despite shorter training durations, the qualitative results remain consistent, with a wide range of buffer compositions significantly improving performance.

**Fine-Tuning.** In multi-task training, it is common to reduce the learning rate for subsequent tasks to fine-tune the network on the next task. To investigate this, we replicate the experiment in Fig. 4a, lowering the learning rate for the second task by a factor of 10. The results align closely with those in Fig. 4a, indicating that our findings are robust and not solely attributable to the specific learning rates used in the original experiments. These results can be found in Fig. 23c.

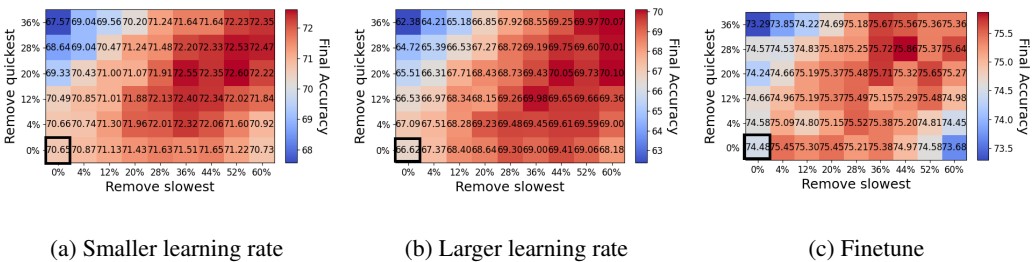

(a) Smaller learning rate      (b) Larger learning rate      (c) Finetune

Figure 23: Repeating Fig. 4a with varying learning rates: (a) doubling the learning rate, (b) halving the learning rate, and (c) keeping the first task's learning rate unchanged while reducing the second task's by a factor of 10. These scenarios, common in continual learning research, yield consistent qualitative results, indicating that our analysis is not dependent on a specific learning rate.

**Regularization.** All ResNet-18 experiments in the main paper were conducted with a small weight decay of 0.0005. To assess the impact of this regularization, we replicate the experiment from Fig. 4a in Fig. 21b where we remove the weight decay entirely. The results remain qualitatively consistent, indicating that the conclusions from our analysis are not sensitive to this specific regularization choice.

**Different splits of CIFAR.** We repeated the experiment from Fig. 4a using a different split of CIFAR-100, where classes were randomly assigned to tasks. The results, shown in Fig. 21c, remain consistent with those in Fig. 4a. Importantly, all task splits in our experiments were across the paper chosen arbitrarily, ensuring unbiased evaluations.

# D STANDARD ERRORS

In the main paper, we omitted the standard errors for visualization porpuses from Figs. 4,5,6 and Table 1. In both cases, these were usually very small and did not affect the qualitative results. For completeness, we report the omitted values in this section. The standard error for Figs. 4,5,6 can be found in Figs. 24,25,26 respectively. The standard errors of Table 1 can be found in Table 2.

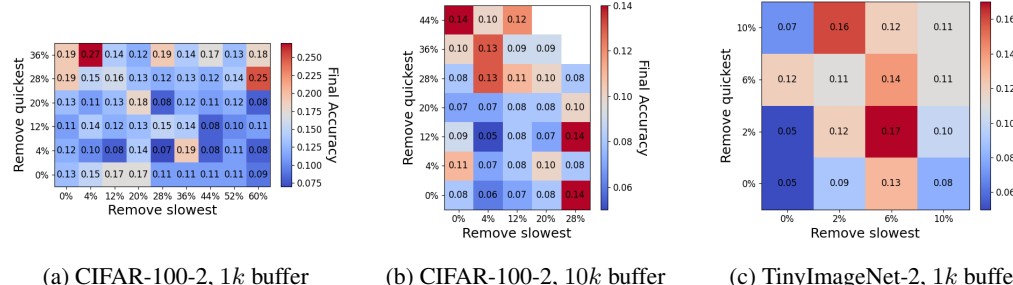

(a) CIFAR-100-2, $1k$ buffer     (b) CIFAR-100-2, $10k$ buffer     (c) TinyImageNet-2, $1k$ buffer

Figure 24: Standard errors for Fig. 4. The error is taken over 10 repetitions in each experiment.

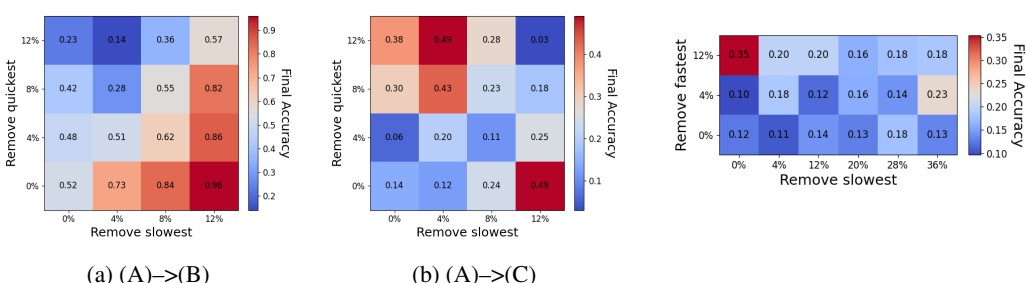

(a) (A)–>(B)     (b) (A)–>(C)

Figure 25: Standard errors for Fig. 5. The error is taken over 10 repetitions in each experiment.

(a) GEM, CIFAR-100-2, 500 buffer

Figure 26: Standard errors for Fig. 6. The error is taken over 10 repetitions in each experiment.

Table 2: Standard error of Table 1.

|  | CIFAR-10-2 | | CIFAR-100-2 | | CIFAR-100-20 | | CIFAR-10-5 | | TinyImageNet-2 | |
| --- | --- | --- | --- | --- | --- | --- | --- | --- | --- | --- |
| Buffer size | $1k$ | $10k$ | $1k$ | $10k$ | $1k$ | $10k$ | $1k$ | $10k$ | $1k$ | $10k$ |
| Random | 0.06 | 0.05 | 0.22 | 0.12 | 0.18 | 0.04 | 0.1 | 0.11 | 0.15 | 0.07 |
| Max entropy | 0.23 | 0.06 | 0.23 | 0.15 | 0.12 | 0.02 | 0.1 | 0.03 | 0.2 | 0.13 |
| IPM | 0.16 | 0.13 | 0.29 | 0.16 | 0.17 | 0.06 | 0.15 | 0.06 | 0.15 | 0.13 |
| GSS | 0.13 | 0.08 | 0.29 | 0.16 | 0.15 | 0.01 | 0.11 | 0.04 | 0.13 | 0.07 |
| Herding | 0.09 | 0.12 | 0.24 | 0.14 | 0.11 | 0.1 | 0.14 | 0.06 | 0.16 | 0.11 |
| Goldilocks | 0.03 | 0.1 | 0.22 | 0.1 | 0.1 | 0.02 | 0.1 | 0.12 | 0.16 | 0.1 |

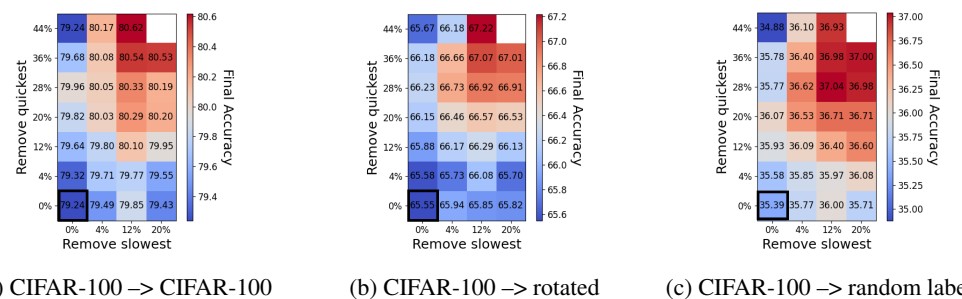

(a) CIFAR-100 –> CIFAR-100        (b) CIFAR-100 –> rotated        (c) CIFAR-100 –> random labels

Figure 28: Comparing replay buffer compositions when training the same original task, followed by different subsequent tasks. Each entry in the matrices denotes the mean final test accuracy of 10 networks, trained with experience replay on 2 tasks, with a buffer size of $10k$. This Figure replicated the same experimental setup as Fig. 5. In (a) we train the models on the 50 first classes of CIFAR-100, followed by the last 50 classes of CIFAR-100. In (b) we train the models on the 50 first classes of CIFAR-100, followed by a rotation classification task on the same examples (see text). In (c) we train the models on the 50 first classes of CIFAR-100, followed by the last 50 classes of CIFAR-100, but with random labels. In all cases, the same buffer compositions of the first task give the same qualitative results, suggesting that the optimal composition of the buffer of the first task is independent of the subsequent task.

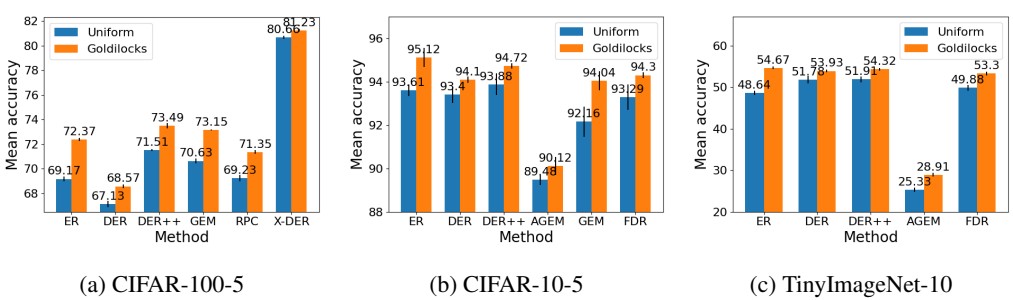

(a) CIFAR-100-5        (b) CIFAR-10-5        (c) TinyImageNet-10

Figure 27: Goldilocks vs. uniform sampling with various continual learning methods. Each bar shows the mean final test accuracy across all tasks for each method, with a uniform sampled buffer (traditionally done) in blue, and Goldilocks in orange. Error bars denote the standard error. (a) CIFAR-100-5 (b) CIFAR-10-5 (c) TinyImageNet-10, all with a buffer of 500 examples. Goldilocks significantly improves performance across all methods and datasets.

# E   INDEPENDENCY OF THE OPTIMAL BUFFER COMPOSITION WHEN CHANGING THE SUBSEQUENT TASKS

We empirically examine how subsequent tasks impact the ideal composition of a replay buffer for a given task. We find that regardless of the similarity or dissimilarity between subsequent tasks and the original task, the optimal replay buffer composition remains largely independent and consistent. To show this, we conduct experiments with three distinct tasks denoted A, B, and C. By training the model on task A and subsequently introducing either task B or task C while keeping the original task unchanged, we analyze the ideal replay buffer composition under varying subsequent tasks.

In Section 3.3, we demonstrate that when selecting tasks A, B, and C from the same dataset, the optimal buffer composition of task A remains unaffected by tasks B or C. We observe consistent quantitative behavior across different *quick* and *slow* parameters. We further explore task variations by setting A as the first 50 classes of CIFAR-100, B as the last 50 classes of CIFAR-100, and C as a rotation classification (Gidaris et al., 2018) task. In the rotation classification task, examples from A are randomly rotated by angles of $\{90°, 180°, 270°\}$, with labels adjusted accordingly. We evaluate the performance of 10 networks trained continuously with different buffer compositions first on task A

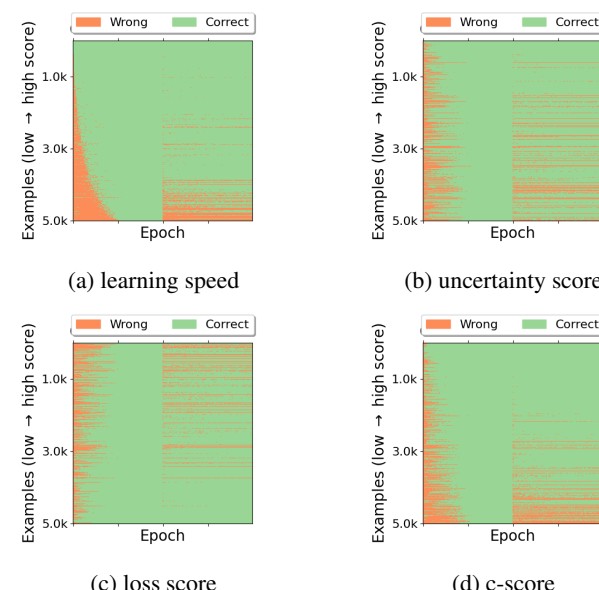

(a) learning speed        (b) uncertainty score

(c) loss score        (d) c-score

Figure 29: Comparison of the extended epoch-wise classification matrix for CIFAR-100-2, sorted by different example-level scoring functions (see App. G). (a) Learning speed (used throughout the paper), (b) Uncertainty measure from Bang et al. (2021), (c) Final loss, (d) c-score from Jiang et al. (2020). While all scores show some correlation with forgetting, as more complex examples tend to be forgotten more, learning speed shows the strongest, suggesting it is most suitable for our analysis.

and then on task B (Fig. 28a), and on task A followed by task C (rotation) (Fig. 28b). Consistent with previous findings, the performance of various buffer compositions remains consistent, independent of the choice of B or C. This property enables us to utilize the rotation task for hyper-parameter grid search without requiring additional label data.

We further modify task C to comprise examples from the last 50 classes of CIFAR-100 but with random labels (Zhang et al., 2021). Training on random labels forces the network to memorize the examples, as no generalization is possible, eliminating any potential overlap between the subsequent tasks. The results of training on task A followed by the random label task are depicted in Fig. 28c. Analogous to the rotation classification scenario, we observe that the same buffer compositions of task A remain effective, further suggesting that the optimal buffer composition of A does not depend on the subsequent task.

## F  GOLDILOCKS WITH DIFFERENT CONTINUAL LEARNING METHODS

Here, we compare Goldilocks to uniform sampling with different continual learning algorithms. We tested several different algorithms on CIFAR-100-5, CIFAR-10-5, and TinyImageNet-10. The results, which extend Fig. 8, are plotted in Fig. 27.

## G  RELATIONSHIP BETWEEN OTHER SCORING METHODS AND THE LEARNING SPEED

To analyze catastrophic forgetting, we focus on its relationship with learning speed (Eq.1). While other metrics exist—such as uncertainty score (Bang et al., 2021), which evaluates classification consistency under augmentations, and c-score (Jiang et al., 2020), which measures expected accuracy on held-out instances – they incur higher computational costs. Using per-example loss as a score is computationally simpler but shows a weak correlation with catastrophic forgetting. As shown in Fig.29, examples sorted by uncertainty and c-score show a moderate correlation with forgetting, while

learning speed provides the strongest correlation. Therefore, we adopt learning speed throughout this work for its computational efficiency and strong alignment with catastrophic forgetting.

