# OpenReview forum: "Forgetting Order of Continual Learning: What is Learned First is Forgotten Last"
_ICLR.cc/2025/Conference — Submitted to ICLR 2025_

### Official Review · Reviewer_m8pC · 2024-10-28

**Soundness:** 4
**Presentation:** 4
**Contribution:** 4
**Rating:** 10
**Confidence:** 4

**Summary:**

The authors claim & show that examples that are learned first (simple examples), are in general not forgotten, while examples that are the hardest are forgotten quickly. They propose a replay sampling method that attempts to counter-balance this phenomenon by replaying only samples that are of medium difficulty.

**Strengths:**

- The authors make an interesting observation that could have strong impact in understanding the learning process of neural networks and improving the replay-based continual learning methods.
- Strong evidence is brought on CIFAR100, using different tools (Figure 2), among which training of multiple networks and consistent observation across these networks that learning speed is strongly correlated with forgetting rate.
- They obtain consistent improvements when applying their sampling method on top of existing methods, and across datasets (CIFAR 100 , CIFAR 10 and TinyImagenet)
- The results are clearly presented using several demonstration tools and the designed method is simple, the ablation of the number of quickly learned samples and slowly learned samples is comprehensive and easy to read.

**Weaknesses:**

- **W1** Maybe a bit more attention could be given to the engineering of class-incremental learning results to make them comparable to the sota one. Right now they are only given on CIFAR100-2 with buffer size of 500. Would be interesting to have them on CIFAR100-10 with bs of 1k or 2k for instance, and maybe applying some anti task-recency bias method or simply probing the representations to show whether the probed representation from the model using the new sampling method is better.

**Questions:**

- **Q1** It is good that results for both CIL and TIL are presented, but for the CIL, they are way less furnished. Would it be possible to have the same than Figure 2 for the CIL in the appendix ?

---

> ### Author Response · Authors · 2024-11-25
> **Response for Reviewer m8pC**
>
> Thank you for your review and positive feedback! We appreciate the recognition of the strengths in our work and the valuable suggestions for improvement. Below, we address your comments and questions:
>
> **W1 + Q1:**
> To address this suggestion, we have significantly expanded the section on CIL in the revised manuscript. Specifically, we now include in Appendix A a repetition of all major figures from the main paper under CIL settings. These include:
>
> * Figure 9 (repeating Figure 2)
> * Figure 10 (repeating Figure 3)
> * Figure 11 (repeating Figure 4)
> * Figure 12 (repeating Figure 6)
> * Figure 13 (repeating Figure 7)
> * Figure 14 (repeating Figure 8)
>
> This expanded analysis shows that, while CIL naturally leads to lower accuracy compared to TIL (as it is inherently more challenging), the qualitative trends and conclusions remain consistent. The results reaffirm that the insights and analysis in the TIL case also hold for CIL.

---

### Official Review · Reviewer_9eZ5 · 2024-10-29

**Soundness:** 2
**Presentation:** 2
**Contribution:** 2
**Rating:** 5
**Confidence:** 5

**Summary:**

In Continual Learning, the methods that have worked best are memory-based. These methods work by sampling a percentage of the training set of each task that is then used in the training of subsequent tasks to ‘remember’ past tasks. In this paper, the authors analyse the best examples to populate the buffer. They analyse the learning speed, showing how it affects performance when sampling from the training set by leaving out the top slower or quickest-to-learn samples. The authors show that items learned quickly are the least forgotten, and conversely, items learned more slowly are the first to be forgotten. With this insight, the authors present a new methodology for populating memory called ‘Goldilocks’. Empirically, the authors show that sampling only from items with an intermediate learning speed can have comparable or better results than current methods for populating memory across different benchmarks.

**Strengths:**

- The authors' motivation for presenting the problem is evident in their approach, which aids in understanding the problem and its relevance.
- An analysis is presented that helps to understand the method before it is presented. Multiple experiments show the usefulness of eliminating the very fast and slow-to-learn examples, to sample only intermediate ones.

**Weaknesses:**

- Despite the authors' thorough analysis, no explanation or intuition is provided as to why medium learning speed items are the most useful for populating memory. It would be good if the authors provided a rationale beyond the empirical results. This rationale could be based on intuition or other work.
- The results shown are limited to a small group of scenarios. The analyses performed are only based on CIFAR10 and CIFAR100 divided into 2 tasks. A better analysis should emphasize a broader set of scenarios and benchmarks to ensure the generalisability of the performance.
    - Other works have shown that performance can change drastically as the number of tasks increases.
    - The analysis shown is with the Task-Incremental learning scenario, I recommend considering the class-incremental scenario as it is a more widely accepted scenario. The authors mention that the analysis is in the Appendix, but I did not find corresponding results.
    - This may affect figures such as Fig2a, where you can see that the forgetting is not as drastic as in class incremental and even a slight increase is seen near epoch 150.
- Although the authors show, both in their analysis and with their method, that the results achieved are better than other alternatives, the benefit is only marginal. Often even less than the standard deviation.
    - During the analysis, the difference is often at most 2%, between all removal combinations slowest/quickest. This shows that the margin of improvement is very slight compared to uniformly populating the memory.
- Some arguments and comments in the paper are difficult to extract from the results.
    - One example is in line 397: 'We find that regardless of the similarity or dissimilarity between subsequent tasks and the original task, the optimal replay buffer composition remains largely independent and consistent'. Nowhere does it show how different or similar the tasks they use are, and they base this only on experiments in CIFAR100.

**Questions:**

- A score called c-score [1] seeks to explain how consistent an example is during training. Can learning speed be related to this score?
- The same order of classes is always used, which may affect the conclusions drawn. Is there a reason for this?
    - Each seed used to run the experiments commonly brings a new class order. This helps to not bias the results to a particular order that may benefit one method over another.
- In line 212, the authors mention using an experience replay strategy that alternates batches of data from the new task and the replay buffer. Why use this and not the standard approach of mixing samples from the current task and the buffer in a 50-50 way?
- Can the learning rate chosen affect the results and conclusions?
    - For example, in fine-tuning, it is recommended to use a small learning rate so as not to modify the old weights significantly.
- Do the authors have results for different CL methods with different strategies to populate the memory? The methods are usually independent of how the data is sampled, so a complete comparison of how much sampling methods affect different memory-based methods can be done.
- I understand using 500 examples for CIFAR10 and CIFAR100, but in TinyImagenet, this means less than 3 elements per class, which can strongly affect the sampling methods used. Do you have experiments with a higher number? It would also be essential to mention the reference to the 'original work' in line 466.

[1] Jiang, Ziheng, et al. "Characterizing Structural Regularities of Labeled Data in Overparameterized Models." International Conference on Machine Learning. PMLR, 2021.

---

> ### Author Response · Authors · 2024-11-25
> **Response for reviewer 9eZ5**
>
> Thank you for your elaborate review. Below, we address each of your points separately.
>
> **Intuition**
>
> Simplicity bias research suggests that neural networks learn examples in order of increasing complexity. At the end of a task, the network performs well on examples up to a certain complexity level. For replay buffers, this implies that:
>
> * Excluding slowly learned (high-complexity) examples makes sense because the network struggled to learn them initially and is unlikely to benefit from replaying them.
> * Excluding quickly learned (low-complexity) examples is also reasonable because these examples provide limited additional value, given the network’s ability to handle more complex tasks.
>
> We added a discussion of this point to Section 3.2.
>
> **Other datasets**
>
> While Figs. 2 and 3 focus on 2-task scenarios in CIFAR-10 and CIFAR-100, the paper also includes results from broader settings. For instance:
> * Fig. 4c: Tiny-ImageNet with 2 tasks
> * Fig. 7: CIFAR-10 with 5 tasks and CIFAR-100 with 20 tasks
> * Fig. 8: CIFAR-100 with 5 tasks and Tiny-ImageNet with 10 tasks
> Although these figures do not explicitly quantify the correlation between learning speed and catastrophic forgetting, the success of Goldilocks in these scenarios implicitly relies on this relationship.
>
> To address your suggestion directly, we have added new results in Appendix B (Fig. 19) and Section 2, validating the correlation from Fig. 2c across diverse datasets and task configurations, including CIFAR-100 with 20 tasks, CIFAR-10 with 5 tasks, and Tiny-ImageNet with 2 tasks finding a stronger correlation.
>
> **Task incremental learning**
>
> Both class-incremental and task-incremental learning are important and widely accepted paradigms in the community. Our choice to focus on task-incremental learning in the main paper was deliberate, as we believe it provides a "cleaner" view of catastrophic forgetting.
>
> That said, we recognize the value of evaluating both scenarios. To address this, we conducted a parallel analysis in the class-incremental setting and included these results in Appendix A of the original manuscript. In the revised manuscript, we expanded the analysis, repeating all Figs 2, 3, 4, 6, 7, 8 with CIL setting, in Figs. 9-14 respectively (App A). A short discussion was added to Sec 2.2, noting that the CIL results mirror those of TIL.
>
> **Marginal benefits and statistical significance**
> We discovered a bug in the code used to generate Fig 24, which contained the standard errors for Fig 4. This was fixed in the revision, showing that all the observed results are statistically significant.
>
> Other than the standard errors themselves, the consistent results across a large range of hyperparameters and experiments also indicate statistical significance. Large standard errors would have resulted in highly erratic behavior, particularly in experiments with closely related hyperparameters. Instead, our findings demonstrate smooth and systematic improvements.
>
> We note that a 2\% improvement in final accuracy is non-trivial in the continual learning domain, and many works report much smaller improvements. Moreover, for scenarios with multiple tasks (Fig 8), the gains are more pronounced, reaching up to 5\%.
>
> **Extracting arguments from the results**
>
> While section 3.3 in the main paper focuses on subclasses of CIFAR-100, we address broader task dissimilarities in App E. We include scenarios where the second task is:
> * Rotated version of the first task, with a different objective
> * Random labels
> Both setups introduce dissimilarity between tasks (see Fig 28 in App E). We have rephrased the relevant text in Section 3.3 to clarify this point.
>
> **c-score:**
>
> Thank you for bringing this reference to our attention. The c-score measures the expected accuracy of individual examples and, like other accuracy-based metrics, shows a strong correlation with learning speed. In Appendix G, we discuss similar accuracy-based scores and their relationship to catastrophic forgetting. Among all these scores, learning speed has the strongest correlation to catastrophic forgetting, and is the cheapest to compute, making it well-suited for our analysis. We added a short discussion and a direct comparison of the c-score to App G and Fig 29d.
>
> **Class order:**
>
> The specific split of classes into tasks did not affect our results. We added to the revised manuscript results with another data-split, in App c, Fig 21c. The results are similar to original ones in Fig.4.
>
> **Experience replay strategy:**
>
> In our experiments, we observed no significant difference between alternating batches and mixing samples from the current task and the replay buffer in a 50-50 ratio. The choice to alternate batches was made purely for code convenience. This point is noted in the revised manuscript for transparency.

---

> > ### Author Response · Authors · 2024-11-25
> > **continued response**
> >
> > **Learning rate:**
> >
> > We conducted additional experiments, repeating the analysis in Fig 4 with:
> > * A different learning rate from the beginning.
> > * A smaller learning rate for subsequent tasks, as you suggested.
> > The results show that the qualitative trends remain consistent in both scenarios, see Appendix C and Fig 23.
> >
> > **Continual learning without uniform sampling:**
> >
> > Our analysis primarily focused on methods employing uniform buffer sampling for two reasons:
> >
> > * Many state-of-the-art (SOTA) methods rely on uniform sampling, allowing for broader and more direct comparisons.
> > * Methods designed for non-uniform sampling are often highly specialized and may fail under alternative sampling strategies like ours or random sampling. Evidence for this is shown in Table 1, where sampling strategies from other methods often underperform compared to random sampling across different CL scenarios.
> >
> > **Tiny ImageNet buffer size:**
> >
> > We acknowledge that using a small buffer for Tiny ImageNet could disproportionately affect certain sampling methods. However, we also evaluated Tiny ImageNet with a significantly larger buffer of 10k examples in Figure 4c. This provides a broader perspective on how buffer size impacts the results and ensures our conclusions are not limited to small buffer scenarios.

---

> > > ### Comment · Reviewer_9eZ5 · 2024-11-27
> > >
> > > I thank the authors for their detailed answers and modifications to the paper.
> > >
> > > **Intuition**
> > >
> > > Although I agree with the small explanation, it would be great to know (or have an intuition) about what makes an example slow (or quick) to learn. For example, one could suggest that because it is a slow-to-learn example, it is necessary to add it to the memory so the model doesn't have to re-learn it after a long process.
> > > What do you mean by "provide limited additional value"?
> > > Some results suggest something else, but it is not entirely intuitive.
> > > For example:
> > > - Are slow-to-learn examples outside of the class distribution (as some previous work suggests)?
> > > - Are quick-to-learn examples more in distribution?
> > >
> > > **Datasets**
> > >
> > > I appreciate the authors' completion of the results on those datasets; however, I still believe that different benchmarks could provide a clearer picture of how this method behaves, giving greater validity to the results shown in the current work.
> > >
> > > **Marginal benefits and statistical significance**
> > >
> > > I understand the consistent improvement over uniform sampling shown in Figure 8. However, even with fixing the bug that measures the Standard error, the difference between other replay buffer sampling methods is not significant in most cases. Compared with the second-best, Goldilock achieved less than a 1% improvement. This may not be sufficient, considering the need to find the percentage of slow and fast elements to eliminate, as suggested by another reviewer, and the difference in performance that can be seen in the figures shown in the paper.
> > >
> > > **Continual learning without uniform sampling**
> > >
> > > I don't entirely agree that "non-uniform samplings are often highly specialized." The main reason people tend to prefer uniform distribution is the low-performance increase of "non-uniform samplings, " which, in my opinion, is also happening in the proposed method.
> > >
> > >
> > > In summary, I agree that the proposed method have potential to contribute to the CL community. However, the paper needs an essential explanation for why removing slow and fast examples helps achieve slightly better performance than other sampling methods. This extra analysis can help the work make a real contribution, not only to the CL area. In addition, understanding the rationale for which elements need to be removed could help eliminate the dependency on both hyper-parameters.
> > > Because of this, I am raising my score but I am still inclined to reject the paper.

---

### Official Review · Reviewer_NS7R · 2024-11-03

**Soundness:** 3
**Presentation:** 3
**Contribution:** 2
**Rating:** 6
**Confidence:** 4

**Summary:**

The paper explores a strategy for selecting examples to include in a replay buffer for continual learning. The main idea is to exclude two sets of examples: those that are learned too easily and those that are difficult to learn, with the aim of improving generalization across a sequence of classification tasks.

**Strengths:**

The authors take this fairly simple idea and run a series of tests.  These experiments cover a range of datasets and settings for the size of the buffer of replayed examples. They also explore two different task orderings and show that the results are consistent across them. Most of the experiments focus on a sequence consisting of just a pair of tasks, but there are some results with a more extensive set of tasks.  The experimentation and reporting of results is clear and fairly complete, especially with the standard error discussion and class incremental results presented in the Appendix.

**Weaknesses:**

The chief weakness is a lack of significance.  The paper is mostly an exploration of whether a type of simplicity bias can be used to guide the selection of examples in the replay buffer. It does not advance a substantive new method or analysis, but seems like a straightforward application of existing ideas.  The results show a consistent but not whopping win for this approach,

A second weakness is a lack of analysis of the types of examples that fit into the too-easy and too-hard categories. Showing that the examples that are learned earlier are forgotten less and those that are learned later are forgotten more is not surprising, as it fits well with various studies such as the simplicity work (as acknowledged by the authors).

As well there is quite a bit of variation across the datasets and experimental conditions, such as buffer size, in terms of the relative performance of different percentages of the too-small and too-fast sets that should be excluded.  There is no analysis of this, which begs the question of how to set these hyperparameters in a new setting.

**Questions:**

I'd recommend that the authors make the method more practically applicable by showing how it can be deployed in a few new settings (e.g., combination of dataset and replay buffer size). One way to address this would be to demonstrate that a small amount of data and experimentation can be used to determine a set of hyperparameters that exhibit strong performance.

One minor question concerns the title, which doesn't quite fit the primary message of the paper.

---

> ### Author Response · Authors · 2024-11-25
> **Response to reviewer NS7R**
>
> Thank you for your review. Below, we address the different points you raised separately:
>
> **Weaknesses — significance and paper's focus**
>
> As noted in the general comment, our work adopts an observational approach, focusing on uncovering behavioral patterns in neural networks rather than introducing a new method or theoretical framework. Specifically, we empirically identify a novel and previously unreported connection between simplicity bias and catastrophic forgetting. While simplicity bias — where networks learn simple examples first — is well-known, we demonstrate that catastrophic forgetting exhibits a "reverse simplicity bias," where complex examples are forgotten before simpler ones.
>
> The perceived lack of "surprise" may stem from the robustness of simplicity bias, which intuitively suggests its relevance to forgetting. As almost any neural network learns simple things first, it is intuitive that an equally robust pattern will occur when such a network forgets. We believe that this robustness enhances the significance of our findings: if forgetting patterns mirror simplicity bias, this suggests a foundational phenomenon that can guide future work in continual learning, leveraging established insights and tools from simplicity bias research. While the "surprise" of a result is subjective, we argue that our contribution lies not in the novelty of simplicity bias itself but in its new application and implications for continual learning.
>
> As for Goldilocks, it serves as a proof of concept to demonstrate how accounting for this forgetting pattern can improve continual learning. Although we do not agree that the improvement is not dramatic (2-4\% of consistent accuracy gain across different scenarios is not easy to achieve), Goldilocks is mainly designed to show that even a relatively simple sampling function based on the connection between simplicity bias and continual learning can improve a large array of continual learning methods, demonstrating the potential for future works to take into account this connection when devising new methods.
>
> Finally, the title reflects the primary focus of the paper: uncovering and understanding the connection between forgetting and catastrophic forgetting. As Goldilocks serves as a practical example, we do not think it should be featured in the title.
>
> **Setting hyperparameters**
>
> The wide range of hyperparameters in our experiments aimed to provide a comprehensive view of how buffer compositions impact forgetting. For practical deployment, we added guidance in Section 3.3 of the revised manuscript to simplify hyperparameter selection for new settings.
>
> In summary:
>
> * A heuristic approach is often sufficient. For instance, setting $q=s=20\%$ performed well across all datasets and scenarios tested.
>
>
> * Adjustments can be made based on prior knowledge of the dataset's complexity:
>
>     * For complex datasets, favor excluding more complex examples $(s>q)$.
>
>     * For simpler datasets, favor excluding simpler examples $(q>s)$.
>
> * For scenarios where computational resources allow, a non-heuristic approach based on auxiliary tasks can further optimize hyperparameters, even with limited data.
>
> We hope these additions address your concerns about practical applicability and encourage you to view this work as a foundation for further exploration of this phenomenon.

---

> > ### Comment · Reviewer_NS7R · 2024-11-27
> >
> > I appreciate the detailed responses to my review as well as the others. These have addressed many of the issues. I also like the more detailed discussion of findings on class-incremental learning.
> >
> > However some of the responses to my review are not adequate. The paper and responses do not contain sufficient analysis of the types of examples that fit into the too-easy and too-hard categories. This is especially important in what you are calling an observational paper -- are there some characteristics of examples that are readily forgotten beyond their learning speed? This could also add some insight into why having relatively fewer of them in the replay buffer may be beneficial. Another reviewer also brought this point up, but I did not see any response to it.
> >
> > Also my point about the title was not that Goldilocks should be in the title but rather that it does not capture the main contribution of the paper. The title focuses on the forgetting order, which is the first of the three contributions highlighted by the authors, and does not address other contributions, such as how this forgetting order should be taken into account in replay to ameliorate forgetting.
> >
> > Nonetheless I will raise my score one point to push it from weak reject to weak accept.

---

> > > ### Author Response · Authors · 2024-12-02
> > >
> > > Thank you very much for your follow-up comments and for raising your score based on our discussion. We greatly appreciate the time and thought you have put into reviewing our work.
> > >
> > > We wanted to kindly note that while you agreed to update your score from weak reject to weak accept, the official review in the system is still marked as 5 ("marginally below the acceptance threshold"). Could you please update the score in the system to reflect your revised assessment of 6? This adjustment would ensure that our discussion is accurately represented.
> > >
> > > ----
> > >
> > > Regarding your additional comments:
> > >
> > > **Example characteristics:**
> > >
> > > We agree that characterizing which examples fall into the too-easy and too-hard categories is an intriguing and important question. Prior work on simplicity bias (e.g., [1], [2], [3]) provides various perspectives on which examples are learned more quickly, with factors such as frequency patterns ([1]), image characteristics in specific contexts ([2]), or the rank of required solution ([3]) influencing learning speed. However, the correlation between learning speed and forgetting is not perfect, suggesting that the characteristics driving forgetting may differ slightly. Due to the nature of continual learning, we also suspect that these characteristics are even more context-dependent than those of simplicity bias. While we think this area is worth deeper exploration, we believe it falls outside the scope of the current work and would make for an excellent subject for future research.
> > >
> > > **Title:**
> > >
> > > Thank you for clarifying your comment. We will consider revising the title in future iterations to better reflect the broader contributions of the work.
> > >
> > >
> > > ----
> > >
> > > [1] Rahaman, Nasim, et al. "On the spectral bias of neural networks." International conference on machine learning. PMLR, 2019.
> > >
> > > [2] Pliushch, Iuliia, et al. "When deep classifiers agree: Analyzing correlations between learning order and image statistics." European conference on computer vision. Cham: Springer Nature Switzerland, 2022.
> > >
> > > [3] Huh, Minyoung, et al. "The low-rank simplicity bias in deep networks." arXiv preprint arXiv:2103.10427 (2021).

---

### Official Review · Reviewer_pNCs · 2024-11-03

**Soundness:** 2
**Presentation:** 2
**Contribution:** 2
**Rating:** 3
**Confidence:** 5

**Summary:**

In this paper, the authors present an empirical study that reveals a strong correlation between catastrophic forgetting and the learning speed of examples. They found that the examples that are learned early in the continual learning process are rarely forgotten, while those learned later are more susceptible to forgetting. Leveraging this finding, they introduced a new replay buffer sampling method - Goldilocks that filters out examples learned too quickly or too slowly, keeping those learned at an intermediate speed. On several low to mid-complexity image classification tasks, they showed the efficacy of their proposed method.

**Strengths:**

Strength:

* The analysis of learning speed and catastrophic forgetting in continual learning is new.
* The authors presented the idea clearly.
* Illustrations and figures - especially the binary classification matrix plots are very useful in understanding the concept of the paper.

**Weaknesses:**

Weaknesses:
* The observed correlation between example learning speed and catastrophic forgetting is empirical, with no theoretical analysis provided, hence of limited significance.
* Empirical analysis provided to establish the correlation is not sufficient. For example, learning dynamics depend on various factors such as learning rate, network architecture, optimizer, regularization etc. One of the major issues with the current paper is that it does not explore these dimensions to establish the correlation between example learning speed and catastrophic forgetting.
* How learning rate for different tasks (initial tasks and later tasks) impact the correlation? If we use a smaller learning rate for later tasks how do forgetting dynamics change? A detailed study is missing here.
* How does the correlation change if plain SGD, Adam, Ada-Grad, etc. optimizers are used?
* The paper only explores ResNet and its smaller variants for the analysis. For other architectures such as transformers, VGG net, etc do the same conclusions stand?
* Gridlock is evaluated on low-to-mid complexity image classification tasks only. Detailed analysis on higher complexity classification tasks on ImageNet is missing.
* As stated in the limitation section, the method does not apply to online CL settings and is only limited to classification tasks.

**Questions:**

See the Weakness section above.

---

> ### Author Response · Authors · 2024-11-25
> **Response for reviewer pNCs**
>
> Thank you very much for your review! Below, we address separately each point in your review:
>
> **Theory**
>
> As noted in the general comment, this paper takes an empirical approach, focusing on behavioral observations to reveal novel insights into neural networks. While we acknowledge that theoretical analysis can strengthen the findings, we believe that empirical evidence also plays a crucial role in advancing understanding, especially in areas like continual learning where theoretical tools may be limited.
>
> **Correlation under different learning hyper-parameters**
>
> Regarding your comments about the empirical analysis, we incorporated your suggestions for additional experiments into the revised manuscript, which we believe improved its quality.
>
> Throughout our empirical study, we explored various datasets and settings, selecting hyperparameters such as optimizers, architectures, and regularization heuristically. Given the extensive hyperparameter space in deep learning, exhaustively testing every result against all possible configurations is infeasible for a single paper. However, we incorporated your suggestions and extended one of our main results (specifically Figure 4) to include the proposed hyperparameter variations. Notably, we found that these variations do not alter the reported correlation: the relationship between learning speed and catastrophic forgetting remains consistent. Moreover, buffer compositions effective in one setting often generalize well to others, further supporting the robustness of our findings.
>
> The new experiments are detailed in Appendix C and referenced in Section 3.2. Specifically, we added:
> * Comparisons across different learning rates, including lowering the learning rates for subsequent tasks. These results can be found in Figure 23
> * Comparisons across different optimizers, including Adam, SGD, and Adagrad, as suggested. These results can be found in Figure 20.
> * Results on a non-ResNet-based architecture, specifically VGG-16, in Figure 21a.
> * Comparisons between training with and without regularization. These results can be found in Figure 21b.
>
> Regarding dataset complexity, our results already include experiments on Tiny ImageNet (Figures 4 and 8), which is commonly considered as challenging as ImageNet. Nevertheless, we plan to include experiments on ImageNet subsets in the camera-ready version, as they take too long to include in the limited time of the rebuttal.
>
> -----
>
> Thank you again for your constructive feedback, which helped us improve our manuscript. We hope our responses have sufficiently addressed your concerns, and we believe the revisions strengthen the paper. Given the mixed reviews, every point is crucial, and we hope you will consider our clarifications in your final evaluation.

---

### Official Review · Reviewer_XrZi · 2024-11-03

**Soundness:** 3
**Presentation:** 3
**Contribution:** 4
**Rating:** 8
**Confidence:** 4

**Summary:**

The paper analyzes the forgetting discrepancies among different examples and provides a theory that the examples that are learned the first and last are the least prone to forgetting. The paper also proposes a practical algorithm for sample selection for the replay buffer where it removes the examples that are learned first or last.

**Strengths:**

- The paper demonstrates simplicity bias in neural networks.
- The paper proposes an effective replay buffer sample selection algorithm that outperforms uniform in many cases and also other subsampling algorithms in some cases.

**Weaknesses:**

- Completeness: Table-1 should also include CIFAR-100-5, CIFAR-100-20 and Tiny-ImageNet.
- Limitation: The conclusion may depend on the training time on each task. For example, if the number of epochs is small, then the hardest to learn examples have not been learned, then it may also need to stay in the replay buffer. The paper has also acknowledged that the method may not be suitable for stream learning in its limitation section. However, it would be better if the paper can give guidance on the number of epochs required for the proposed method to work well.
- Hyperparameters: The algorithm may rely on selecting hyperparameters (e.g. s and q) for removing the slowest and fastest examples. And it might be unclear how that parameter varies across different datasets. If choosing a hyperparameter repetitive experiments, then it may defeat the premise of continual learning.

**Questions:**

- I wonder if the authors can provide experiments on other datasets, and show how hyperparameters will vary across different datasets.

---

> ### Author Response · Authors · 2024-11-25
> **Response for Reviewer XrZi**
>
> Thank you very much for your review!
>
> Addressing the weaknesses and questions you raised:
>
> **Completeness:**
>
> We agree that extending Table 1 to include larger datasets such as CIFAR-100-5, CIFAR-100-20, and Tiny-ImageNet would strengthen the manuscript and enhance its completeness. However, we were unable to conduct these additional experiments during the rebuttal period due to the computational demands of certain baselines (e.g., GSS and IPM) and the modifications required to adapt their code for multi-task settings. We will incorporate these datasets into Table 1 for the camera-ready version.
>
> **Limitation:**
>
> Training Goldilocks with a limited number of training epochs presents two potential challenges. First, because the *learning speed* (Eq. (1)) depends on the number of epochs, its accuracy diminishes as the number of epochs decreases. Second, as you noted, if the networks train for too few epochs, they may not converge, leading to instances where examples are incorrectly classified as "slowly learned" due to insufficient training.
>
> In the revised manuscript, we address these concerns by presenting additional experiments (Appendix C + Figure 22). These experiments show that the same buffer compositions remain effective even with significantly reduced training iterations. Although there is some noise introduced by the less accurate *learning speed* in these cases, the qualitative results remain consistent. On the other hand, for scenarios like streaming data, where the network may not converge adequately, the *learning speed* simply can not capture well enough how fast the model learns certain examples, and Goldilocks requires some different complexity measurement to work well (which is beyond the scope of our work).
>
> As recommended, we have incorporated these findings into the limitations section (Section 4), and we discuss the results from Figure 22 in Appendix C and Section 3.2 to provide practical guidance on the appropriate number of epochs needed for the method to perform optimally.
>
>
> **Hyperparameters:**
>
> Figs. 4-6 present a wide range of hyperparameters $q$ and $s$ to demonstrate their impact comprehensively. These results indicate that a broad selection of $q$ and $s$ values can effectively enhance learning, highlighting the ease of hyperparameter selection for new datasets. For instance, in all tested settings and datasets, simply setting $s=q=20\%$ significantly improved performance.
>
> These straightforward choices can be further refined based on dataset characteristics: for challenging datasets, setting $q < s$ may be beneficial, as slowly learned examples are less likely to contribute to learning and can be removed more aggressively. Conversely, for simpler datasets, choosing $s < q$ may be preferable. Importantly, these heuristic adjustments require no additional computation and align with the premise of continual learning.
>
> While many continual learning studies provide hyperparameters without justification, we opted to include explanations to guide practitioners. For scenarios with additional computational resources, we also propose a systematic approach for hyperparameter tuning, which we have described in Section 3.3. Additionally, we added to the revised manuscript, at the end of section 3.3, a paragraph explaining these points, and guiding how to pick in practice hyper-parameters when encountering new data, which we hope will help guide future readers.
>
> **Question:**
>
> In the paper, we present results on multiple datasets, including CIFAR-100-2, CIFAR-100-20, CIFAR-10-2, CIFAR-10-5, TinyImageNet-2, and TinyImageNet-10. Due to the limited time available during the rebuttal period, we were unable to include experiments on datasets significantly different from those already evaluated. However, we added an explanation on how to pick hyper-parameters at the end of Section 3.3. Additionally, we plan to incorporate additional datasets for the camera-ready submission. Moreover, in this revised manuscript, we have included results for an alternative split of CIFAR-100 into two tasks, with classes split into each task randomly. These results are provided in Appendix C, Figure 21c.

---

> > ### Comment · Reviewer_XrZi · 2024-11-25
> >
> > I thank the authors for their response.
> >
> > - I still believe presenting a full suite of results on all datasets in Table 1 is necessary and the computational demands of baseline shouldn't be a top concern.
> >
> > - I appreciate the additional results in the Appendix. It does seem that the optimal values of s and q shifts over the number of training epochs.
> >
> > - It is true that some continual learning methods also rely on hyperparameters, but I was comparing it to random uniform sampling. Also, I don't think Figure 4-6 gives adequate guidance on proper selection of s and q. It seems like the optimal values are also very dependent on the dataset and the buffer size. The authors gave some qualitative comments on "dataset characteristics" but I don't think they are backed by empirical results.

---

> > > ### Author Response · Authors · 2024-11-27
> > >
> > > Thank you for the swift response!
> > >
> > > **Updated Table 1:**
> > >
> > > We fully agree that presenting Table 1 with all the datasets is crucial for the completeness of the work. Based on your suggestion, we have now included in the revised manuscript Table 1 results for all the datasets you suggested, including CIFAR-100-20, CIFAR-10-5, and TinyImageNet. These results confirm that while other sampling methods occasionally succeed or fail depending on the dataset and buffer size, Goldilocks consistently performs well across all tested scenarios. We hope these additions address your concerns about the completeness of our evaluation.
> > >
> > > **Hyperparameter Selection and Practical Guidance:**
> > >
> > > We appreciate your detailed comments on hyperparameter selection. While our heatmaps in Figures 4-6 were intended to provide an extensive overview of the relationship between learning speed and forgetting, and not to guide the selection of $q$ and $s$, we understand that this broader analysis may give the impression of complexity. While the maximal value in each heatmap may vary across experiments, in practice, Goldilocks does not require the *optimal* hyperparameters to outperform uniform sampling. Across all datasets, a very wide range of $q$ and $s$ values leads to improved performance compared to random sampling, and using any of these hyper-parameters will result in a good performance. For example, in the "training epochs" experiments you referred to in Figure 22, any s value between 4\% and 60\% achieves better performance than random sampling, with corresponding q values spanning a broad range, sometimes up to 36 different percentage points. Notably, while the *optimal* values across the experiments in Fig. 22  differ, the range of good hyper-parameters remains very similar.
> > >
> > > To give guidance on how to choose hyper-parameters in practice, there is a paragraph at the end of section 3.3. The suggested method is ultimately heuristic, as the range of good hyper-parameters is so wide, that simple heuristics often get satisfactory results off the shelf, without any additional computing. However, for practitioners with additional resources, we also describe a principled tuning method at the end of Section 3.3, which requires no additional data and performs well empirically. This balance between practicality and flexibility makes Goldilocks applicable while staying true to the constraints of continual learning.
> > >
> > >
> > > **Goldilocks as a Proof of Concept:**
> > >
> > > It is important to emphasize that Goldilocks is primarily a proof of concept demonstrating that continual learning methods can be improved by accounting for which examples are prone to forgetting — which can be predicted before training on a new task. While Goldilocks achieves strong empirical results and can be applied in practice, we envision that future methods could leverage these underlying principles to achieve even greater performance with more sophisticated approaches.
> > >
> > > We hope these clarifications and updates address your concerns and help contextualize the contributions and practicality of Goldilocks. Thank you again for your comments, which helped improve the manuscript.

---

### Author Response · Authors · 2024-11-25
**General comment for the reviewers**

We thank all the reviewers for the constructive reviews and the time and effort they took in reviewing our paper. The diverse range of scores (10, 8, 5, 3, 3) highlights polarized views on our paper. We believe this polarization stems from its unorthodox focus on behavioral analysis rather than conventional methods or theory, and we would like to provide additional context for our approach.

In this paper, we take an observational approach, observing a novel connection between simplicity bias and catastrophic forgetting. While simplicity bias -- where neural networks learn simpler examples before complex ones -- is well-known, we find that in catastrophic forgetting there is a "reverse simplicity bias," where complex examples are forgotten before simpler ones.

Our contributions are threefold:
* Observing and measuring: We uncover this reverse simplicity bias and define tools to quantify it.

* Exploration: We systematically test the phenomenon across diverse scenarios, identifying factors that influence it.

* Practical application: We demonstrate its utility in continual learning by introducing a sampling strategy (Goldilocks) that leverages this insight, improving multiple methods across varied settings.

This type of behavioral investigation, which is focused on observing a phenomenon rather than suggesting a new method or conducting a theoretical study, is inspired by approaches common in neuroscience and psychology, where systems too complex for full theoretical analysis are studied empirically to gain insights from observed behavior. Similarly, as neural networks remain challenging to analyze fully mathematically, we argue that behavioral studies can offer valuable understanding.

We encourage reviewers to evaluate the paper from this perspective. Beyond the sampling method, we ask that you consider whether the observed connection between simplicity bias and catastrophic forgetting is both novel and significant, whether it enhances our understanding of catastrophic forgetting, and whether it has the potential to inspire future research.

We hope this clarification helps reviewers see the broader value of our work beyond the introduction of a new sampling method, and reconsider its contribution to the field.

---

### Meta-Review · Area_Chair_JZHV · 2024-12-25

**Metareview:**

This paper turned out to be a tricky one for me because of the variation in the scores given by different reviewers. While reviewers pNCs, NS7R, and 9eZ5 do not seem to be supporting the acceptance of the paper, reviewers XrZi and m8pC have given very high scores.

I personally read the paper carefully and discussed with the reviewers to ensure fair assessment. Unfortunately, after discussions and reading the reviews carefully, I am recommending rejection of this work in its current form (few reasons mentioned below).


---
The authors investigate the impact of sampling the so-called mid-learned examples on continual learning (CL), they call the underlying sampling method Goldilocks.

Sampling a small subset of samples to store as episodes is important in reducing forgetting in CL and is an important aspect to carefully look at in any storage/compute constrained CL formulation. However, I also believe that such proposals when based purely on intuitions and empirical observations demand thorough investigation.

Reasons why I think the paper should be rejected (most of them have already been mentioned by the reviewers as well)

- The main intuition that different examples are easy/hard to learn and a models' performance on them vary during training is not new (e.g., [1]) and has been the basis of several works including say the renowned focal loss paper. Therefore, I see little novelty when it comes to providing new intuitions. Having said that, using this to propose an effective sampling strategy for CL does have merit and I appreciate the reviewers for connecting the dots here.

- However, I believe that to ensure that the intuition really works well empirically, the set of experiments provided seem to be very limited in scope. They do provide some promising results but aren't enough to accept the claims made in the paper. For example, I would have preferred seeing experiments in (1) both online and offline settings (both settings have formulations that rely on episodic memory); (2) task incremental and class incremental with several classes/task and at least 15-20 tasks to see the true effect of CL; (3) large scale set-ups (imagenet, iNaturalist at the very least); (4) set-ups where a pre-trained backbones are used (e.g., continual fine-tuning of CLIP, or RanPAC type set-ups), _perhaps in this one there isn't much difference between fast and slow learned samples due to rich representations we already obtain from pretrained models_? I think investigating all these aspects would be crucial to provide the right intuition on where one should expect Goldilocks to work. Since the paper is primarily empirical, the study must be done exhaustively across a variety of experiments to justify the arguments made.

- Perhaps, examples learned slowly are more important to keep in the episodic buffer and the ones learned fast are being learned fast due to spurious correlation (Fig 1, the Bees examples are all with yellow background). A similar comment was also made by Reviewer 9eZ5 as well. Discussion along these lines would be highly valuable for the paper.

- The improvements also seem to be marginal given that an extra forward pass is needed every epoch to compute the importance score. I also find the claim in Fig 5 to be too bold given its empirically tested only on two setting A-B and A-C.

I would like to mention that while going through the reviews I realised that the **authors significantly worked during the rebuttal period**, added new experiments etc., and I truly appreciate their effort towards this. I"m sure these efforts will eventually contribute towards making the paper much stronger.

[1] An Empirical Study of Example Forgetting during Deep Neural Network Learning, Toneva et.al., 2019 [2] RanPAC: Random Projections and Pre-trained Models for Continual Learning [3] Fine-tuning can cripple your foundation model; preserving features may be the solution

**Additional Comments On Reviewer Discussion:**

- The concerns raised by the reviewers were mainly regarding (1) lack of proper experiments (small datasets, use of old architectures); (2) lack of novelty behind the main idea (and lack of theoretical justifications); (3) lack of proper analysis (dependence on learning rate, architecture etc.); (4) restricted scenarios (e.g, did not perform experiments on both online and offline settings, task and class-incremental) etc.
- Reviewers greatly appreciated the active participation of the authors during the rebuttal. Authors did provide several new results such as (CIFAR-100-20, CIFAR-10-5, and TinyImageNet), ablation (effect of different optimizer, learning rate etc.) and arguments that were compelling.
- However, there was no unanimous agreement towards the acceptance of this work primarily because of the reasons mentioned above. Since the work is empirical in nature, the experiments must be rigorous.

---

### Decision · Program_Chairs · 2025-01-22

Reject